# Identifying therapeutic targets by combining transcriptional data with ordinal clinical measurements

Leila Pirhaji[1], Pamela Milani[1], Simona Dalin [1], Brook T. Wassie[1], Denise E. Dunn[2], Robert J. Fenster[3,4,5], Julian Avila-Pacheco[6], Paul Greengard[4], Clary B. Clish [6], Myriam Heiman[3,4,7], Donald C. Lo[2] & Ernest Fraenkel [1,6]

The immense and growing repositories of transcriptional data may contain critical insights for developing new therapies. Current approaches to mining these data largely rely on binary classifications of disease vs. control, and are not able to incorporate measures of disease severity. We report an analytical approach to integrate ordinal clinical information with transcriptomics. We apply this method to public data for a large cohort of Huntington's disease patients and controls, identifying and prioritizing phenotype-associated genes. We verify the role of a high-ranked gene in dysregulation of sphingolipid metabolism in the disease and demonstrate that inhibiting the enzyme, sphingosine-1-phosphate lyase 1 (SPL), has neuroprotective effects in Huntington's disease models. Finally, we show that one consequence of inhibiting SPL is intracellular inhibition of histone deacetylases, thus linking our observations in sphingolipid metabolism to a well-characterized Huntington's disease pathway. Our approach is easily applied to any data with ordinal clinical measurements, and may deepen our understanding of disease processes.

[1] Department of Biological Engineering, Massachusetts Institute of Technology, 77 Massachusetts Avenue, Cambridge, Massachusetts 02139, USA. [2] Center for Drug Discovery, Department of Neurobiology, Duke University Medical Center, 303 Research Drive, Durham, North Carolina 27710, USA. [3] Picower Institute for Learning and Memory, 43 Vassar St, Cambridge, Massachusetts 02139, USA. [4] Laboratory of Cellular and Molecular Neuroscience, The Rockefeller University, 1230 York Ave, New York, New York 10065, USA. [5] McLean Hospital, 115 Mill Street, Belmont, Massachusetts 02478, USA. [6] Broad Institute, 415 Main St, Cambridge, Massachusetts 02142, USA. [7] MIT Department of Brain and Cognitive Sciences, 77 Massachusetts Avenue, Cambridge, Massachusetts 02139, USA. Leila Pirhaji and Pamela Milani contributed equally to this work  Correspondence and requests for materials should be addressed to E.F. (email: fraenkel-admin@mit.edu)

Transcriptional profiling technologies are now so routine that databases such as the NCBI Gene Expression Omnibus (GEO) and ArrayExpress each contain more than 1.5 million samples. This growth has led to a significant need for computational methods to infer biological insights from these data[1]. Methods have been developed to identify clusters of biological samples with specific pattern of expression, enabling molecular stratification of diseases such as cancer[2]. Expression data have also facilitated discovery of biomarkers[3], identification of signatures corresponding to disease progression, and profiles resulting from cellular perturbations[4]. Nevertheless, identification and prioritization of gene subsets that influence disease phenotypes remain challenging.

The search for disease-associated genes and biomarkers relies on the discovery of statistical links between gene expression and disease phenotype. In most methods, clinical metrics are treated as binary data[5] (e.g., disease vs. control). However, in many cases, even the most basic clinical data provide a richer description of the disease process. Rating scales such as the Tumor, Node, Metastasis staging of tumors[6], Glasgow Outcome Score related to brain injuries and Clinical Dementia Rating[7] provide a measure of the degree of severity or progression of a disease that are typically excluded from analyses. Systematic integration of these ordinal clinical metrics with gene expression data may lead to identifying a subset of the genes that play a critical role in disease progression. Once experimentally validated, these genes could be important candidates for therapeutic targets.

However, existing approaches for discovering genes associated with ordinal clinical categories, such as multi-way ANOVA analysis and the Kruskal–Wallis test, do not take into account the ordinal relationship between the categories. These tests have been widely used for comparing multiple phenotypic categories[8], but these methods consider the categories independently. On the other hand, approaches that are based on correlation analysis[9] consider the relative ranking value of ordinal categories. However, clinical phenotypes have a qualitative nature, and a severity score of four does not represent twice the severity of a score of two.

To develop an approach that can take advantage of information on the severity of the disease, we analyzed gene expression data from the brains of patients who suffered from Huntington's disease (HD), a genetic neurological disorder caused by a CAG repeat expansion in the gene encoding the huntingtin protein. Transcriptional dysregulation is one of the earliest and most fundamental events in disease pathogenesis[10], and has been reported in multiple HD models[11], making it likely that some expression changes could cause later pathology. In addition, the neurophysiology of HD is well understood. Neurons in the striatum and other brain regions atrophy, and these losses are strongly associated with the clinical manifestation of HD[12]. Patients who died of HD can be classified in five categories, called Vonsattel grades, based on the severity and pattern of neurodegeneration[13]. We reasoned that combining the qualitative neurohistology represented by the Vonsattel grades with transcriptomic data from patient brains could be used to identify a subset of genes whose transcriptional dysregulation leads to neuropathological changes.

Using a systematic, data-driven approach, we analyzed the relationship between the Vonsattel grade and gene expression data in a large cohort of HD patients and controls. By adapting a principled statistical method, we identified *SGPL1* (a key regulator of sphingolipid metabolism) as a gene whose transcriptional dysregulation is strongly associated with progressive neurodegeneration in HD. We then confirmed the importance of the expression changes through a meta-analysis of gene expression in five distinct HD models. These data confirmed that genes involved in the sphingolipid pathway are dysregulated in HD models. We then validated the role of *SGPL1* as a potential therapeutic target in well-established models of the disease using

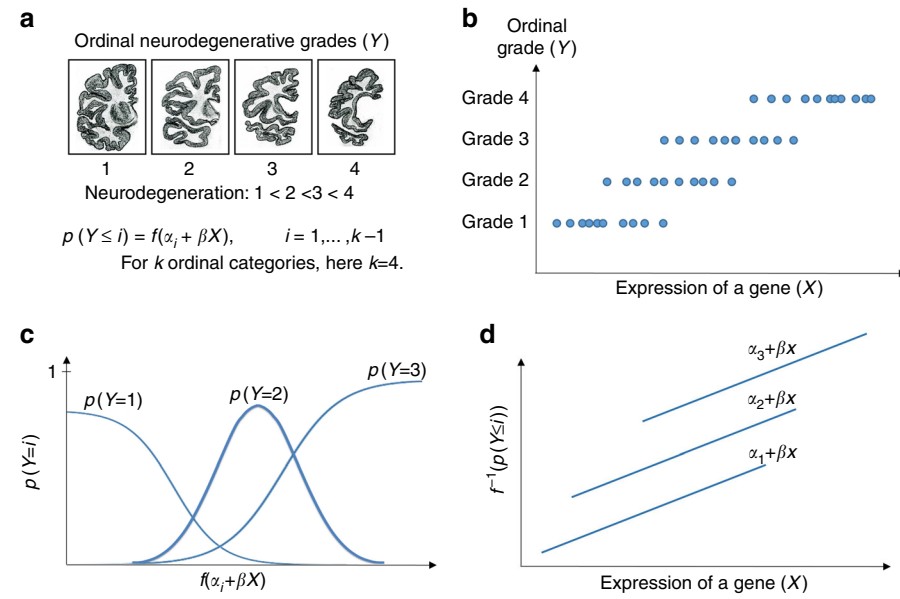

**Fig. 1** Combining transcriptomic data with ordinal clinical information. **a** Schematic representation of postmortem brain tissues, which are categorized in four groups based on their degree of neurodegeneration. The formula shows an ordinal regression model in which the expression of a gene explains the degree of neuronal loss. This model is implemented using the proportional odds assumption, where the slope of the fitted lines ($\beta$) is considered equal for all categories. **b** In this plot, the X axis shows the expression of a gene, while the Y axis shows neurodegenerative grades. Each dot represents the expression of a gene from a sample with neurodegenerative grade y. **c** The plot shows the ordinal regression model, in which a linear function of the expression of a gene is related to the probability of an ordinal category. **d** In this plot, the expression of a gene is represented in the X axis, while the Y axis shows a function of the cumulative probability of ordinal categories. The parallel lines display fitted lines by implementing the proportional odds assumption

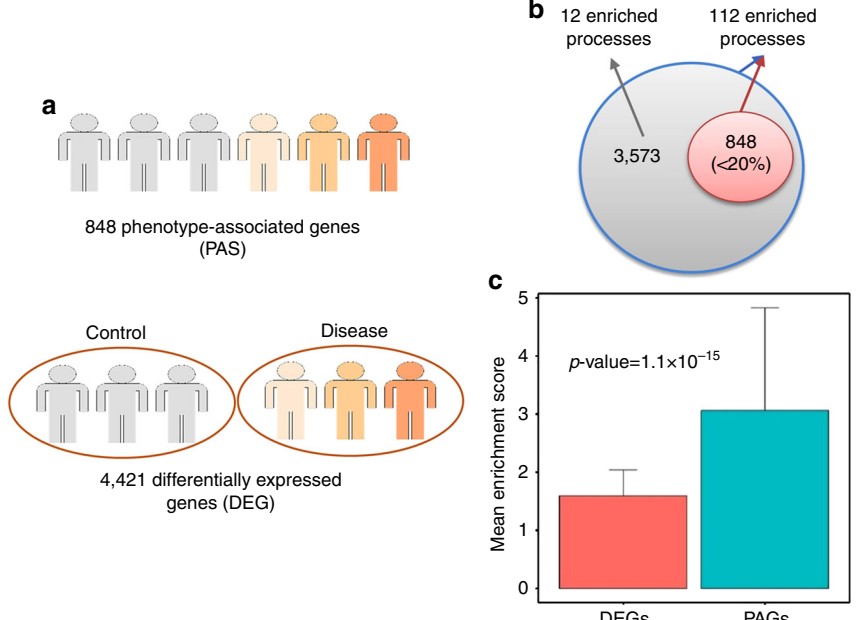

**Fig. 2** Comparisons of PAGs with DEGs. **a** 848 genes are identified by considering the severity of the disease, while 4421 genes (corrected Limma $p < 0.001$) are identified when disease samples with different grades of severity are considered as one group. **b** PAGs are a small fraction (<20%) of DEGs, and they are enriched in the same 112 biological processes as DEGs (Hypergeometric test, corrected $p < 0.01$). The Figure further shows that the subgroup of DEGs not identified by the ordinal regression are only enriched in a few biological processes. **c** The *bar plot* shows that the mean enrichment score for PAGs is significantly higher than DEGs (two-tail *t*-test, $p = 1.1 \times 10^{-15}$). The *horizontal bars* show the mean enrichment score of biological processes, and the *error bars* show the standard deviation

knock-down and chemical inhibition of the enzyme. These experiments also pointed to potential mechanisms of action by which targeting *SGPL1* exerts cell-protective effects. Our approach for systematic integrative analysis of transcriptomic data and ordinal clinical information has provided new insight into HD, and can be applied broadly to the identification of novel therapeutic targets in other diseases.

## Results

**Ordinal regression model to link clinical and transcriptomic data.** To distinguish and prioritize genes whose transcriptional dysregulation is associated with pathogenic effects, we jointly analyzed transcriptomic data with an ordinal, qualitative measure of the clinical state of postmortem brain tissues from HD patients. We first compiled gene expression data from the caudate nucleus of 38 HD patients and 32 neurologically normal controls that we obtained from the NCBI GEO database[11]. Clinical metrics corresponding to disease phenotype for these samples are publicly available (detailed in Methods). Specifically, we categorized the patients based on the reported severity of neurodegeneration.

Next, we sought to identify and rank genes that are linked strongly to the severity of HD neurodegeneration (Fig. 1a). For this purpose, we adapted a statistical model known as ordinal regression to integrate real-valued expression data with the Vonsattel grade, which represents macroscopic and microscopic changes in the striatum and is significantly associated with the clinical symptoms of HD[13]. The ordinal regression model makes no assumptions about the relative quantitative value of the scale. In this regression model, the values of the response variable have an ordinal relationship, e.g., low, medium and high. Using this model, we sought to identify genes whose expression is associated with the degree of neurodegeneration (Fig. 1b, c).

To fit the ordinal regression models, we employed the proportional odds assumption for which the slope of the fitted lines is equal among odds-ratios of categories (Fig. 1d). The use of the proportional odds assumption decreases the number of fitted parameters, and thus reduces the bias towards input data. One important advantage of the ordinal regression model implemented with the proportional odds assumption is the ability to overcome the challenges posed by gene expression data of tissues with advanced neurodegeneration. Stage four caudate nucleus is likely to have far fewer neurons than stage one. As a result, simply looking for genes that have large differences in expression between early and late stages may be misleading, as the results will be dominated by changes in cell type. By contrast, our approach requires that the genes be altered even in the early stage of HD, before significant neuronal loss, and it requires that the expression of the gene must consistently increase or decrease with HD progression.

Using this approach, we identified 848 genes (termed phenotype-associated genes, PAGs, Supplementary Data 1) whose transcriptional dysregulation was significantly associated with neuropathological severity (two-tailed *z*-test, *p*-value of all the fitted parameters <1e-6). The magnitude of the β parameter, which is fitted for each gene, correlates to the rate by which the expression of a gene is altered with the increase of the grade in HD neurodegeneration. We identified 226 consistently upregulated PAGs, and 622 consistently downregulated PAGs due to progressive neurodegeneration. Gene ontology (GO) analyses assessed the enrichment of upregulated genes in biological processes, including RNA metabolic processes; downregulated genes were enriched in neuronal processes such as synaptic transmission and ion transport (Supplementary Table 1). We then ranked PAGs based on the slope of the fitted lines.

Next, we compared the results of our ordinal regression model to the ones obtained from differential analysis of gene expression data of control and disease categories, disregarding their severity (Fig. 2a). We first inferred 4421 differentially expressed genes (DEGs, corrected Limma test, $p < 0.001$, detailed in Methods)

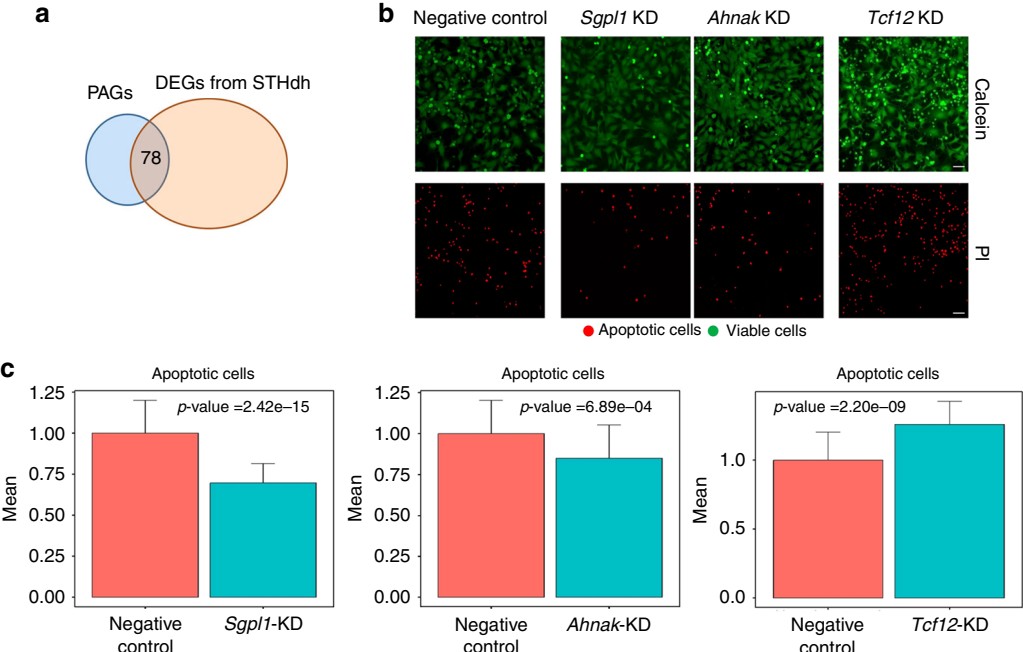

**Fig. 3** High-ranked upregulated PAGs are modifiers of cellular viability. **a** 78 overlapping genes were identified between human PAGs and STHdh DEGs. **b** Knocking-down high-ranked upregulated genes significantly alters apoptosis in STHdh Q111 cells compared to cells transfected with negative control siRNA. Calcein (*green*) stains viable cells, while propidium iodide (PI—*red*) is a marker for late apoptotic cells. *Scale bar*=10 μm. **c** The number of apoptotic cells is represented as fold changes normalized to the control. Two-tailed *t*-test was performed to calculate statistical significance. The *bar* shows the mean value and the *error bars* indicate the standard deviation. Three independent experiments were performed, with 15 replicates each

between control and HD samples (detailed in Methods), which were enriched in 186 biological processes (Hypergeometric test, corrected $p < 0.01$). In comparison, PAGs were a small subset of these DEGs (<20%), and they were enriched in 112 of these processes (Fig. 2b). Interestingly, the DEGs that were not identified by ordinal regression were enriched in only 12 biological processes (Fig. 2b, Hypergeometric test, corrected $p < 0.01$). Notably, the enrichment scores (defined in Methods) for these processes were significantly higher for PAGs than DEGs (Fig. 2c, $p = 1.1 \times 10^{-15}$, two-tailed *t*-test unless specified from now on). Additionally, PAGs had higher enrichment scores in known dysregulated biological processes in HD[10], including synaptic transmission, neurotransmitter transport and secretion and calcium transport (Supplementary Fig. 1). Therefore, our method can select and prioritize a subset of DEGs with higher enrichment for HD-related processes.

**High-ranked phenotype associated genes have roles in HD**. As the slope of the fitted lines indicates the rate by which the expression of a gene changes with progressive neurodegeneration, we hypothesized that genes with larger slopes would be likely to have important roles in the disease. Indeed, the top two genes were Bcl-2-like protein 11 (*BCL2L11*, Supplementary Fig. 2a) and specificity protein 1(*SP1*, Supplementary Fig. 2b), both of which have known roles in HD. *BCL2L11* is a member of the Bcl-2 protein family. The members of this family are involved in mitochondrial apoptosis[14], mitochondrial morphogenesis[15], and metabolism[16]. The upregulation of BimEL, which is the most common isoform transcribed by *BCL2L11* in brain neurons[17], has been shown in several HD models[18]. BimEL upregulation has been further associated with the toxicity of the mutated HD gene[14]. The gene ranked second, *SP1*, encodes a transcription factor that has been well-studied in association with HD pathogenesis[19]. The SP1 protein regulates the expression of genes involved in processes such as cell growth, apoptosis,

differentiation, and immune responses[20]. Although there is no consensus on the exact role of the SP1 transcription factor in HD pathogenesis, several studies have shown that SP1 is upregulated in HD models, and its suppression has protective effects[21].

We next sought to determine whether our approach could identify novel disease-associated genes. For this purpose, we used the well-established STHdh cell line model of HD. STHdh cells are mouse striatal progenitors that express either the wild-type huntingtin gene, with 7 CAG repeats (STHdh Q7), or a mutated form with 111 CAG repeats (STHdh Q111)[22]. We first identified differentially expressed genes between wild-type and disease samples using previously published RNA-sequencing data from these cell lines[23]. We then identified 78 genes that were differentially expressed in STHdh cell lines and whose human homologs were identified in the ordinal regression model (Fig. 3a).

We validated the role of high-ranked genes by performing small interfering RNA-mediated (siRNA) knock-down experiments (Fig. 3b, c). We first selected four high-ranked upregulated genes for these experiments: sphingosine-1-phosphate lyase 1 (*Sgpl1*), AHNAK nucleoprotein (*Ahnak*), transcription factor 12 (*Tcf12*) and tensin 1 (*Tns1*, Supplementary Fig. 3). For this purpose, we transfected STHdh Q111 cells with either siRNAs specifically directed against the transcripts of these selected genes or siRNA negative control for 48 h. We assessed the silencing efficiency by real-time qPCR (Supplementary Fig. 4). We then carried out cell viability experiments after exposing the cells to serum starvation for 24 h. We chose this stress condition because STHdh Q111 cells are known to be more sensitive to serum starvation than STHdh Q7 cells due to the toxicity of the mutated huntingtin protein (mHtt); in fact, the expression of mHtt was associated with enhanced caspase activation[24] and decreased ATP levels[25] in serum-deprived striatal cells. Our results showed three of these high-ranked genes (75%) are modulators of cellular viability (Fig. 3b, c). While knocking-down *Sgpl1* and

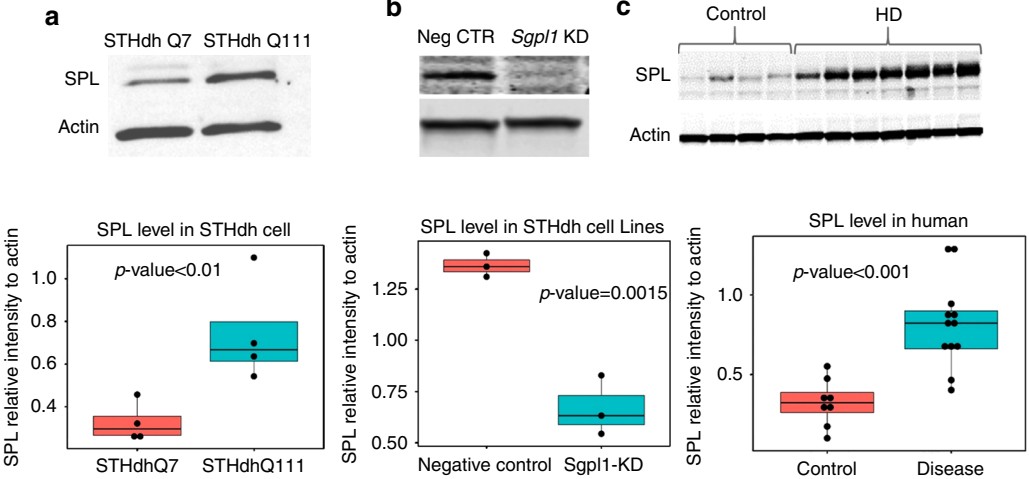

**Fig. 4** SPL enzyme is significantly upregulated in HD. **a** Western blot results indicate that SPL levels are significantly higher in STHdh Q111 cells compared to STHdh Q7 cells ($p < 0.01$, two-tailed $t$-test, 4 replicates). **b** Western blot experiments show a significant decrease in the $Sgpl1$-encoded protein after the knock-down experiments ($p = 0.0015$, two-tailed $t$-test, 3 replicates). The $plot$ shows the levels of $Sgpl$-1 encoded protein. siRNA-mediated knock-down experiments were performed to validate the specificity of the SPL antibody (Abcam, ab56183): STHdh Q111 cells treated with siRNA against $Sgpl1$ ($Sgpl1$ KD) showed a drop in the signal compared to STHdh Q111 cells treated with negative control siRNA (Neg CTR). **c** Western blot analyses reveal that SPL enzyme levels are significantly upregulated ($p < 0.001$, two-tailed $t$-test, 8 control, and 12 HD samples) in postmortem cortical tissues of HD brain compared to controls. The $boxplots$ represent the distribution of the ratios of SPL to actin levels, in which the first and the third quartiles of the data are displayed by the $top$ and the $bottoms$ of respective $boxes$ and the median (second quartile value) is show as a $horizontal line$ within the $box$

$Ahnak$ significantly decreased the toxicity of mHtt in diseased cells ($p = 2.42 \times 10^{-15}$ and $p = 6.89 \times 10^{-4}$, respectively), $Tcf12$ knock-down significantly increased its toxicity ($p = 2.20 \times 10^{-9}$, Fig. 3b, c), which indicates the potential compensatory role of this gene. We did not observe changes in cell viability after $Tns1$ silencing (Supplementary Fig. 5).

**Dysregulated sphingosine metabolism in HD.** We investigated the potential modulatory role of $SGPL1$ in HD pathogenesis, since it is the highest ranked gene among the 78 overlapping genes between PAGs and STHdh DEGs. $SGPL1$ encodes the sphingosine-1-phosphate lyase 1 (SPL) enzyme, which is involved in sphingolipid metabolism[26]. We first measured SPL expression in STHdh cells and human samples by western blot (WB) analysis. The results showed that the level of SPL was increased in STHdh Q111 compared to STHdh Q7 cells ($p < 0.01$, Fig. 4a and Supplementary Fig. 7a). siRNA-mediated knock-down experiments validated the specificity of the antibody raised against mouse SPL protein (Fig. 4b and Supplementary Fig. 7b). We further performed WB experiments on human postmortem cortical brain tissue. Information about these samples is provided in the Supplementary Table 2. Even though the most striking HD-related neuropathological alterations are found in striatum, we decided to test SPL expression in cortical brain tissue for the following reason: cerebral cortex displays pathological features of HD[27], but it exhibits less dramatic neuronal loss than striatum as determined by flow cytometry counts of NeuN-postive cell fraction of HD and control samples[28]. Therefore, this tissue allowed us to validate the ability of the proportional odds model in detecting HD-related genes independent of significant variations in cell-type composition. In line with our findings from the ordinal regression model, we observed a significant increase in SPL level in HD patients ($p < 0.001$, Fig. 4c and Supplementary Fig. 7c).

Since SPL is one of the key modulators of the sphingolipid pathways[29], we investigated the levels of several sphingosine bases in R6/2 mouse model of HD. Using lipid extraction followed by mass spectrometry experiments, we measured

these lipids from the striatum of 6-week-old R6/2 mice on a hybrid C57BL/6×CBA (B6/CBA) background. First, our results showed a significant decrease in the levels of the substrate of SPL, d20:1 sphingosine-1-phosphate (S1P, $p = 0.03$, Supplementary Fig. 6a) in R6/2 compared to wild-type mice. Additionally, we showed a significant decrease in the total levels of sphingosine bases ($p = 0.05$, Supplementary Fig. 6f), as well as several sphingolipids, including d18:0 sphinganine-1-phosphate ($p = 0.009$, Supplementary Fig. 6b), d20:0 sphinganine-1-phosphate ($p = 0.01$, Supplementary Fig. 6c), d18:0 sphinganine ($p = 0.03$, Supplementary Fig. 6d), and d20:1 sphingosine ($p = 0.028$, Supplementary Fig. 6e).

Interestingly, we demonstrated that the levels of S1P decrease significantly with the progression of HD. We first measured S1P levels in the striatum of 6-week-old and 22-week-old R6/2 mice on a pure C57Bl6/J background. These mice show a milder HD phenotype than R6/2 animals on a mixed B6/CBA background, thus allowing us to monitor the changes in S1P levels for an extended period of time[30]. We found that d18:1 S1P levels are significantly decreased in 6-week-old R6/2 mice compared to the corresponding controls ($p = 0.03$, Fig. 5a). In addition to sphingolipids with a chain length of 18 carbons, which include the main components of cellular sphingolipids[31], we detected changes in sphingolipids with a chain length of 20 carbons. Interestingly, the detection of these sphingolipids has been reported in brain and central nervous systems[31], and their dysregulation has been associated with neurodegeneration[32]. We found that the levels of d20:1 S1P and total phosphorylated sphingoid bases are significantly decreased in both 6-week (Fig. 5b, c respectively) and 22-week (Fig. 5d, e respectively) R6/2 mice compared to the corresponding controls. Furthermore, we discovered the decrease in d20:1 S1P levels is significantly associated with the age of the R6/2 mice and disease progression ($R = -0.83$, $p < 0.0098$, Fig. 5f).

Additionally, we determined that the expression of several genes involved in the sphingolipid pathway is dysregulated in multiple HD models by performing a meta-analyses of gene expression data from five HD models including R6/2, R6/1,

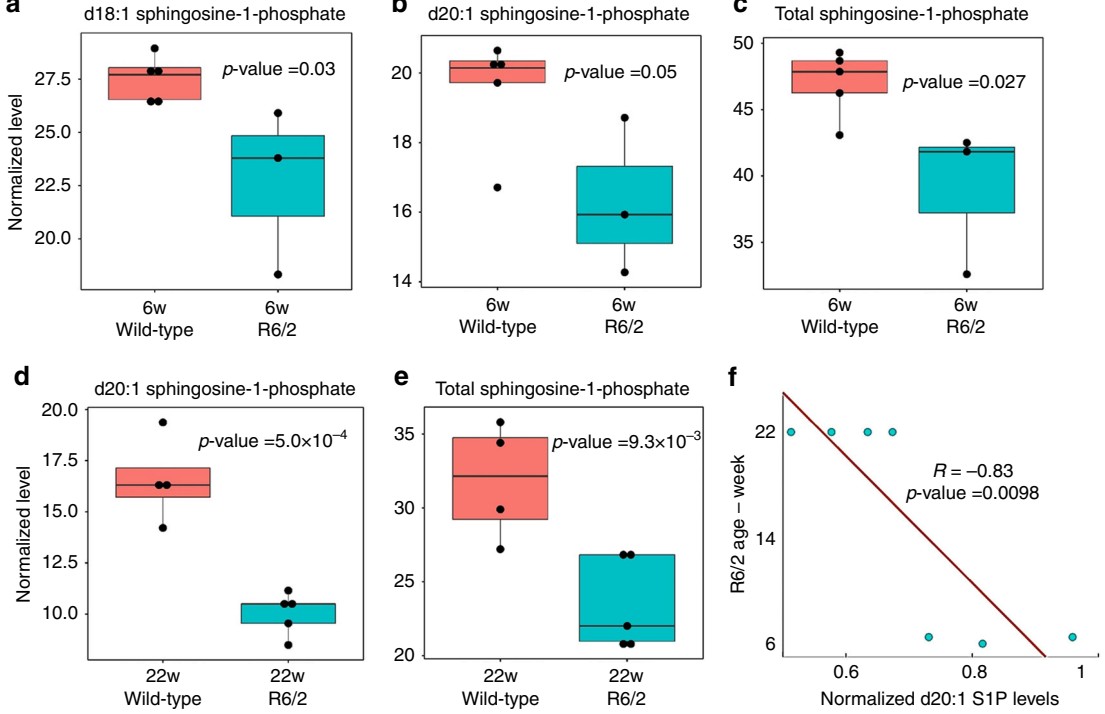

**Fig. 5** Decreased levels of phosphorylated sphingoid bases in R6/2 mice. The *box plots* show the levels of phosphorylated sphingoid bases in 6-week-old and 22-week-old R6/2 mice and corresponding control animals. d18:1 S1P ($p = 0.03$, (**a**)), d20:1 S1P ($p = 0.05$, (**b**)), and total S1P ($p = 0.027$, (**c**)) levels are significantly decreased in 6-week-old R6/2 mice compared to controls. Additionally, d20:1 S1P ($p = 5.0 \times 10^{-4}$, (**d**)) and total S1P ($p = 9.3 \times 10^{-3}$, (**e**)) levels are significantly decreased in 22-week-old R6/2 mice compared to their corresponding wild-type animals. Two-tailed *t*-tests were performed to calculate statistical significance. Each *boxplot* shows the distribution of the levels of phosphorylated sphingoid bases. The *top* and the *bottom* of the *boxplots* show first and third quartiles of each data set, and *middle line* show the medium. Statistical significant is calculated using two-tailed *t*-test. **f** The S1P levels are significantly anti-correlated with the age of R6/2 mice ($R = -0.83$, two-tailed *t*-test, $p = 0.0098$). The *dots* show the normalized S1P levels in 6-week-old R6/2 and 22-week-old R6/2 mice, in which the S1P levels are normalized by the mean of S1P levels in their age-matched controls. The *line* shows a fitted linear regression model, predicting the age of the animals based on the S1P levels

YAC128, human postmortem tissues, and the STHdh cell line. Expression data from these models were obtained from published data deposited at the NCBI GEO database. Initially, we identified 33 human and mouse homolog genes that are involved in the sphingolipid pathway from the KEGG database. Then, we inferred that 14 ($\geq 42\%$) of the genes involved in this pathway were dysregulated in at least one of these HD models (Supplementary Table 3). Collectively, our results indicate the importance of dysregulation of SPL and sphingolipid metabolism to HD pathogenesis, and the potential therapeutic benefits of targeting this pathway.

**SPL inhibition exerts neuroprotective effects**. To assess the therapeutic potential of targeting the SPL enzyme, we tested the effect of inhibiting its activity on cell viability. To this purpose, we added a well-known SPL inhibitor, 4-deoxypyridoxine (DOP)[33], to the culture medium of STHdh Q7 and STHdh Q111 cells in the absence of serum for 24 h. Since the SPL enzyme lyses the S1P lipid, we first evaluated the levels of intracellular S1P after the DOP treatment using liquid chromatography–mass spectrometry experiments. We observed a statistically significant increase in S1P levels after the treatment with 4 mM DOP ($p = 5.03 \times 10^{-5}$, Fig. 6a). We subsequently measured cell death by high-content imaging. Consistent with previous evidence from the literature, STHdh Q111 cells showed higher sensitivity to serum starvation than STHdh Q7 cells (mean cell death: 30.3% and 5.9%, respectively; $p = 1.74 \times 10^{-26}$, Fig. 6b, c). Treatment with DOP significantly reduced apoptosis in serum-deprived STHdh Q111 cells (mean cell death: 24.7%, $p = 2.4 \times 10^{-4}$, Fig. 6b, c),

while only a slight change in viability was observed for STHdh Q7 cells after DOP treatment (mean cell death: 5.2%; $p = 0.022$, Fig. 6b, c).

We further showed that inhibiting SPL exerts neuroprotective effects in an ex vivo model of HD consisting of the biolistic transfection of rat corticostriatal brain slices with a DNA construct derived from the human mutant HD allele. The advantage of this tissue-contextual phenotypic platform over in vitro HD cellular models is that explanted tissues maintain the cytoarchitecture of the brain regions, including glial–neuronal interactions, thus mimicking the multicellular environment present in vivo. These brain slices were co-transfected with a construct encoding the yellow fluorescent protein (YFP) and a construct expressing the mutated exon-1 of *huntingtin* gene harboring 73 CAG repeats (Httex1-Q73). The YFP marker was used to quantify the medium spiny neurons (MSNs) and assess their viability as described by Reinhart et al.[34] and Crittenden et al.[35]. Over the course of 4 days, ~50% of Htt-transfected striatal neurons degenerated. Inhibition of SPL by 0.4 and 4 mM DOP showed a significant increase in the number of healthy MSNs ($p < 0.05$ and $p < 0.01$, respectively, ANOVA followed by Dunnett's post hoc comparison test, Fig. 7).

**SPL inhibition modulates histone acetylation**. We decided to examine the effects of S1P inhibition on the epigenome based on recent studies demonstrating the role of S1P in epigenetic regulation and the fact that epigenetic events are associated with neuronal dysfunctions[36]. S1P directly inhibits the activity of histone deacetylases (HDAC1 and HDAC2), which leads to

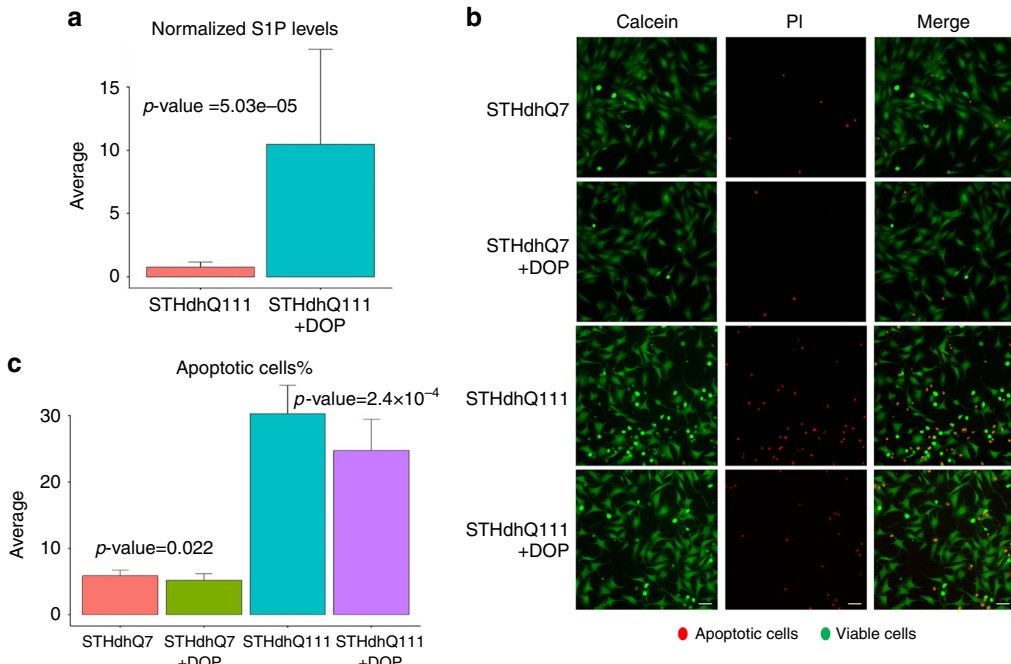

**Fig. 6** Inhibiting the SPL enzyme increases S1P levels and has protective effects in the STHdh cell line model of HD. **a** Untargeted metabolomic measurements show a significant increase in the S1P levels after treatment with DOP ($p = 5.03\text{e-}5$, 14 replicates for untreated samples and 15 replicates for DOP treated samples). **b**, **c** SPL inhibition by DOP significantly decreases apoptosis in STHdhQ111 cells. The fold-change of decrease in cell death is significantly higher in STHdh Q111 cells compared to STHdh Q7 cells. Calcein (*green*) stains viable cells, while propidium iodide (PI—*red*) is a marker for late apoptotic cells. *Scale bar*=10 μm. The *bar plots* show the average levels and the error bar represents the standard deviation in the data. Three independent experiments were performed, with at least 7 replicates each. Two-tailed *t*-test are used for calculating the *p*-values

increased acetylation of specific histone residues, particularly the lysine 9 of histone H3 (H3K9ac)[37]. Alterations in histone acetylation levels have been reported in various neurodegenerative diseases such as HD[38]. In fact, histone H3 and H4 hypo-acetylation has been reported in several HD models[38] including R6/2 transgenic mice. Notably, histone deacetylase inhibitors have been demonstrated to exert neuroprotective effects[39]. Previous investigations have shown that the balance between H3K9ac and H3K9 methylation is affected in HD models such as R6/2 and 82Q mice, and that correcting this ratio improves HD phenotype[40]. Since H3K9ac plays an important role in neuronal functions and S1P can inhibit HDACs, we measured the effect of S1P on H3K9ac in STHdh cell lines.

Chemical inhibition of the SPL enzyme by DOP led to a significant increase in overall levels of H3K9ac (Fig. 8a and Supplementary Fig. 7); the fold change of this increase was significantly higher in treated STHdhQ111 cells compared to treated STHdhQ7 ($p = 0.025$). Furthermore, we confirmed that the incubation of nuclear extracts from STHdhQ111 cells with S1P inhibited HDAC activity ($p = 0.008$, Fig. 8b). Consistent with this result, we showed that treating STHdhQ111 cells with DOP also significantly decreased HDAC activity ($p = 0.01$, Fig. 8c).

**Protective mechanisms associated with increase in H3K9ac.** We next determined the potential downstream effects of the increase in H3K9ac by analyzing transcriptomic and H3K9ac ChIP-sequencing data. For this purpose, we first performed RNA-sequencing experiments in serum-deprived STHdh Q111 cells treated with either DOP or vehicle (Methods). We compared the results with those obtained from serum-deprived STHdh Q7 cells. We identified 3097 differentially expressed genes after DOP treatment of STHdh Q111 cells (using Cuffdiff analysis, corrected $p < 0.01$). Among these genes, we identified a cluster of genes whose expression is downregulated after DOP

treatment and corrected toward STHdh Q7 levels (cluster A, Fig. 9). Cluster A genes are involved in biological processes such as stress response, interferon-beta and immune response (Supplementary Table 4). Similarly, we detected two clusters (cluster B and C, Fig. 9) whose expression levels are increased after DOP treatment and corrected toward STHdh Q7 levels. The genes in cluster B and C are significantly enriched in the regulation of axonogenesis and neurogenesis, Ras and Rho signal transduction, and regulation of neuronal projection development (Supplementary Table 4).

Next, we combined these transcriptomic data with ChIP-seq data for H3K9ac in order to identify and rank genes that are likely to be regulated by an increase in H3K9ac levels (Fig. 10a). We first identified 291 genes that were not only significantly upregulated (using Cuffdiff analysis, corrected $p < 0.01$) after DOP treatment, but also had at least one H3K9ac peak within 10 kb of their promoters (Fig. 10b). We then quantified the mediated effect of H3K9ac score on the gene expression fold-change by calculating the interaction effect (IE) statistic (Methods). We identified 146 genes with an IE score higher than the average. These genes were enriched in components of cytoskeleton, neuronal part and cell body, and cell projection (hyper geometric test, corrected $p < 0.01$, Supplementary Table 5) as well as in biological functions such as protein, ion, cytoskeletal protein bindings and Ras guanyl-nucleotide exchange factor activity (Supplementary Table 5). Collectively, these results show that SPL inhibition and the subsequent increase in H3K9ac activate biological functions involved in neuronal processes.

**Discussion**

We have demonstrated a principled statistical method for identifying genes whose expression is associated with ordinal clinical metrics, and we used this method to elicit novel biological insight from previously published data. Transcriptomic profiling

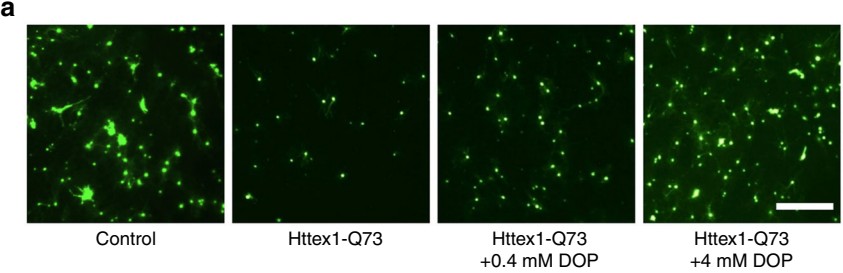

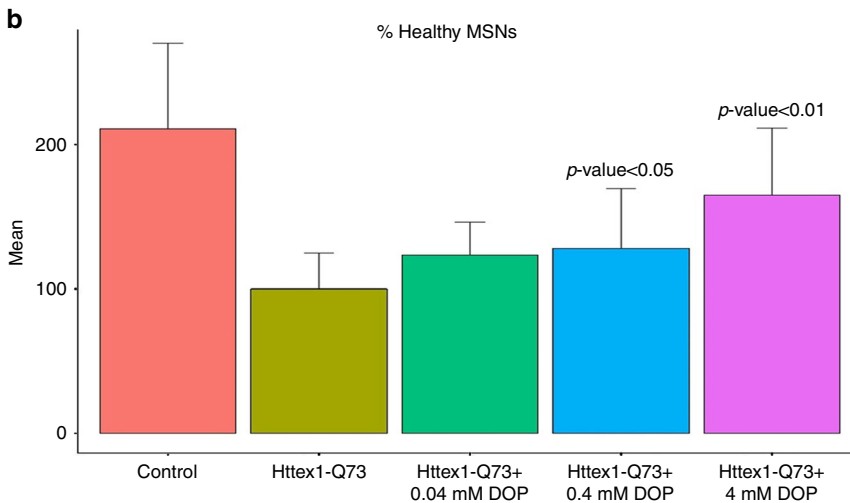

**Fig. 7** Inhibiting SPL has neuroprotective effects in brain slice. **a** Example fluorescence photomicrograph of live brain slices biolistically transfected with YFP to provide a marker for the quantification of healthy MSNs. The micrographs show: MSNs expressing YFP in a control brain slice ("Control"), and in brain slices co-transfected with Httex1-Q73 construct and treated with either vehicle ("Httex1-Q73") or different concentrations of DOP ("Httex1-Q73 + 0.4 mM DOP", "Httex1-Q73 + 4 mM DOP"). *Scale bar*=400 μm. **b** Treating coronal brain slices with DOP significantly increases the number of detected MSNs. Graphs show the numbers of healthy YFP-positive MSNs per striatal region in each brain slice averaged across 3 independent experiments and normalized to the Httex1-Q73 negative control condition set to 100%. Statistical significance was evaluated using ANOVA followed by Dunnett's post hoc comparison test at the 0.05 confidence levels. The *bar plots* show the average levels and the *error bar* represents the standard deviation in the data

is now ubiquitous, and has been used to detect many genes associated with phenotypes in humans, model organisms and other settings. However, existing methods for analyzing gene expression cannot be easily adapted to incorporate clinical rankings of disease severity[5]. These qualitative measures derive from expert knowledge and are used throughout medicine to categorize the severity of symptoms across the spectrum of human maladies. To enable leverage such data, we developed an analytical approach and used it to analyze gene expression data and ordinal clinical information from HD patients and corresponding controls. Our approach identified 848 phenotype-associated genes. These genes show changes in expression that begin at the earliest stage of the disease, when there are no macroscopic changes to the brain, and consistently increase or decrease with disease progression. The categories of genes identified by this analysis were highly consistent with prior literature on HD, and the top-ranked genes were well-studied in the context of this disease.

We carried out experiments to determine whether the changes in the levels of the PAGs were merely correlated with disease progression or can play a causal role. To that end, we knocked-down several top ranked genes and found that 75% of the tested genes altered viability in a cell-based model of HD. To explore the potential for this method to uncover novel disease-related genes, we studied *SGPL1*, one of the top-ranked genes, in detail. We demonstrated that *SGPL1* has the potential

to be a therapeutic target in HD. *SGPL1* encodes the SPL enzyme, a key regulator of sphingolipid metabolism[26]. Sphingolipids are abundant in neuronal cells and maintaining balanced concentrations of sphingolipids is essential for proper neuronal functions. Enzymes involved in the sphingolipid pathway are dysregulated in many neurodegenerative diseases including Alzheimer's disease (AD)[41], amyotrophic lateral sclerosis[42], and HIV-dementia[43]. In particular, previous studies showed the upregulation of the SPL enzyme and downregulation of S1P in AD brains compared to controls[44]. A few studies have specifically examined this pathway in the context of HD. In Pirhaji et al.[45], we showed the dysregulation of sphingolipid metabolism in a cellular model of HD, and downregulation of complex sphingolipids including gangliosides has been shown in HD[46]. Furthermore, treatment with ganglioside GM1 has been shown to reduce the toxicity of mutant huntingtin[47]. Here, we showed that the levels of several sphingolipids such as phosphorylated sphingoid bases are significantly decreased in R6/2 mouse model of HD compared to controls. Additionally, our meta-analyses of gene expression data from five HD models determined that more than 42% of the genes involved in the sphingolipid pathway are dysregulated. We also observed upregulation of the SPL enzyme in human postmortem cortical brain tissue from HD patients compared to neurologically normal controls. Collectively, our results provide evidence of dysregulation of sphingolipid metabolism in HD.

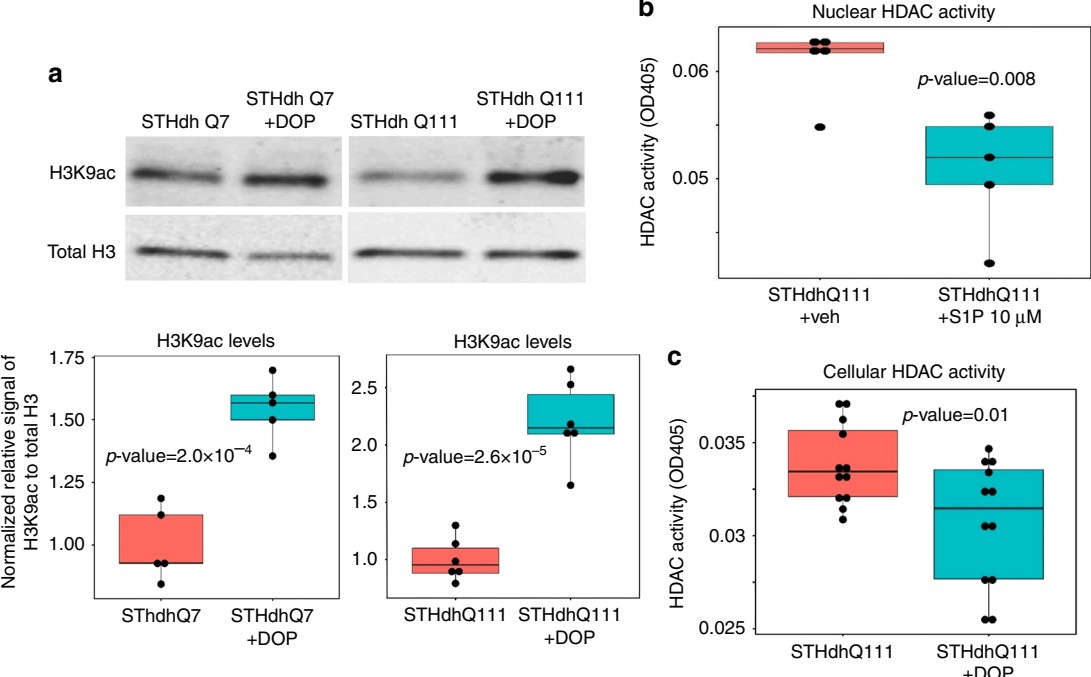

**Fig. 8** Inhibiting the SPL enzyme and increasing S1P levels enhance H3K9 acetylation and decrease HDAC activity. **a** Western blot experiments show a significant increase in H3K9ac levels in STHdh Q111 cells ($p = 2.0 \times 10^{-4}$, two-tailed $t$-test) and STHdh Q7 cells ($p = 2.6 \times 10^{-5}$, two-tailed $t$-test) after treatment with DOP compared to untreated cells. **b** Incubation of nuclear extracts from STHdh Q111 cells with 10 µM S1P diminishes HDAC activity ($p = 0.008$, two-tailed $t$-test). HDAC activity is expressed as optical density at 405 nm (OD405). **c** Treating STHdh Q111 cells with 4 mM DOP significantly decreases HDAC activity (OD405) ($p = 0.01$, two-tailed $t$-test). The boxplots show the distribution of the data, while each *dot* shows a data point corresponding to a biological replicate. The first and the third quartiles of the data are represented by the *top* and the *bottoms* of respective *boxes* and the median (second quartile value) is displayed as a *horizontal line* within the *box*

In addition to demonstrating that inhibiting the SPL enzyme significantly increased survival of cells with mutant huntingtin in the STHdh cell line model of HD, we also found a survival advantage from SPL inhibition in coronal brain slices transfected with mutated huntingtin. Furthermore, we showed that the inhibition of the SPL enzyme and the consequent increase in S1P levels exerted an effect on the epigenome, through changes in the levels of acetylation of H3K9. S1P is an important bioactive molecule that activates intracellular and extracellular signaling pathways leading to proliferation and anti-apoptotic effects[48]. Although the extracellular activities of S1P are well known[49], the intracellular role of S1P in epigenetic regulation was only discovered recently. S1P directly inhibits HDAC1/2 activities, which leads to increased levels of specific classes of histone acetylation, including H3K9[37]. H3K9 acetylation is essential for neuronal function and exerts an important role in learning and memory[36]. Additionally, HDAC inhibition and the subsequent increased acetylation, including H3K9ac, ameliorate HD symptoms[39]. Here, we demonstrated that one potential protective mechanism of action of SPL inhibition is the modulation of HDAC activities and consequent increase in H3K9ac levels.

In conclusion, we have demonstrated that the combination of transcriptomic data with ordinal clinical information advances our understanding of biological processes in disease progression and can lead to the discovery of novel genes with potential therapeutic roles. Considering the exponential increases in publicly accessible transcriptional data, we envision that our systematic approach will have broad applications in biological research in human diseases.

## Methods

**Gene expression data**. We obtained gene expression data for human, mouse and cell line models of HD from the NCBI Gene Expression Omnibus (GEO) database[50]. The human gene expression compendium includes transcriptional data of postmortem striatal tissues for 38 HD patients and 32 unaffected controls, (NCBI GEO entry GSE3790)[11]. In addition to gene expression data, the Vonsattel grade representing the severity of neurodegeneration is available for each patient. We compiled mouse gene expression from the following sources: striata of 12-week R6/2 mice (NCBI GEO entries GSE9803 and GSE9804)[51], striata of 22-month CHL2 mice (NCBI GEO entry GSE10202)[51], striata of 3-month and 18-month HdhQ92 mice (NCBI GEO entry GSE7958)[51], the brains of 18-week, 22-week, and 27-week R6/1 mice (NCBI GEO entry GSE3621)[52], and the striata of 12-month and 24-month YAC128 mice (NCBI GEO entry GSE19677)[53], and corresponding controls for each model. This compendium includes 19 101 genes and 74 arrays. We further obtained transcriptional data for the STHdh cell line model of HD using the NCBI GEO database accession number GSE43433[23].

To analyze microarrays from human and mouse, we applied a two-step normalization process. First, expression data of each microarray chip was normalized using the robust multi-array average method[54] with the R Bioconductor package[55]. In the second step, a quantile normalization was used on the entire gene expression compendium to scale the distributions of expression arrays[56]. Each gene was assigned the average of expression levels for its corresponding probes. Next, we identified differentially expressed genes between each HD model and corresponding controls using the Limma R package[57]. Human homologs of differentially expressed genes in mouse models were further identified using the NCBI HomoloGene database[50].

GO enrichment assessment was performed using GOrilla software[58]. The enrichment score for each GO term is calculated as the ratio of the frequency of the genes associated with a GO term from the target gene list compared to that from the background gene list.

**Ordinal regression model**. Ordinal regression models are useful when more than two categorical responses are present as an ordered series, e.g., high, medium, and low[59]. For $k+1$ ordered categories, an ordinal regression model has k equations and

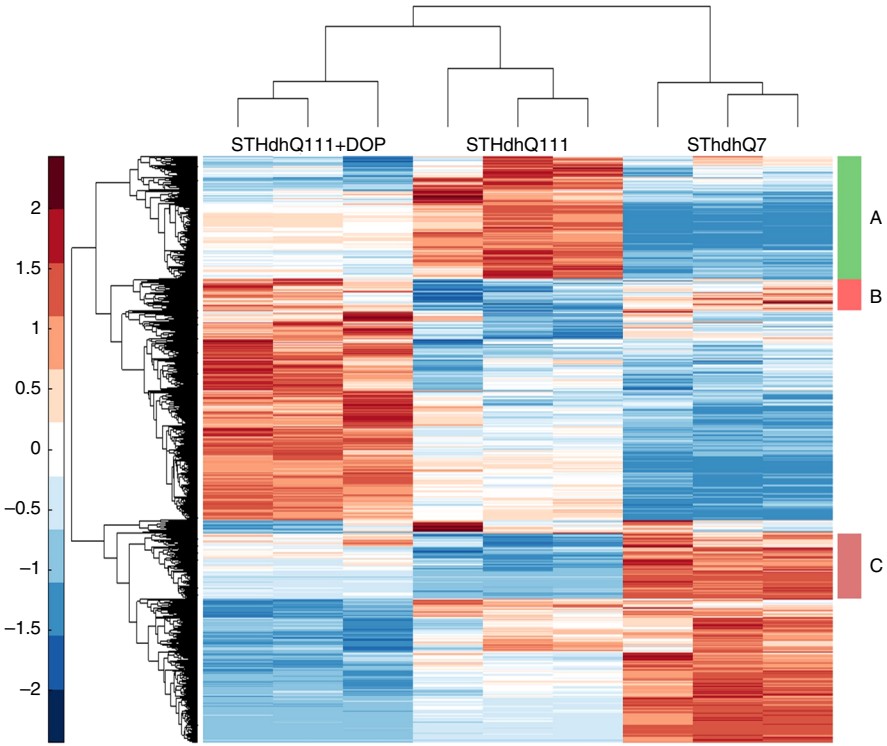

**Fig. 9** Differentially expressed genes after treatment with DOP. Hierarchical clustering of gene expression data from differentially expressed genes in STHdh Q7, STHdh Q111, and STHdh Q111 cells after treatment with 4 mM DOP. Cluster A represents the genes whose expression levels are downregulated after DOP treatment, and are closer to STHdh Q7 expression levels. Clusters B and C show the genes whose expression levels are increased after treatment with 4 mM DOP and are corrected toward STHdh Q7 expression levels

defined as[60]:

$$\text{odds}\,(Y \leq i) = \frac{P(Y \leq i)}{1 - P(Y \leq i)}$$

$$\text{odds}(Y \leq i) = f(\alpha_i + \beta_i X), \; i = 1, \; \ldots, \; k$$

To reduce the number of free parameters, we applied the commonly used proportional odds assumption, in which the $\beta$ parameter is the same for all odds-ratios. Thus, the proportional odds assumption requires determining $k$ intercept values ($\alpha_i$) and one value for the slope ($\beta$)). The proportional odds model for $k+1$ ordinal response categories is:

$$\text{odds}(Y \leq i) = f(\alpha_i + \beta X), \; i = 1, \; \ldots, \; k$$

To fit this model, we applied the R ordinal package. We implemented the probit link function, which is the inverse of the cumulative distribution function for the Gaussian distribution[61], since we assume there exists an underlying latent normal distribution in the response variable. The p-values for inferred parameters from this ordinal regression model were calculated using the two-tailed z-test for the null hypothesis, which consists of a coefficient that is equal to zero.

Next, we applied the ordinal regression model to the human expression compendium and identified genes whose expression levels are explained by the progression of neurodegeneration. Our human gene expression compendium was composed of expression data for 18 898 genes from 32 unaffected controls and 38 HD patients. Clinical information of these patients are classified into ordinal categories based on the grade of neurodegeneration (i.e., Vonsattel grade). Here, we classified these samples into four ordinal categories (Supplementary Table 6), and fitted an ordinal regression model for each gene.

**Cell culture**. The conditionally immortalized murine striatal progenitors expressing either wild type (STHdh Q7) or mutant Htt (STHdh Q111) were purchased from Coriell (CH00097 and CH00095, respectively) and grown as described in Trettel et al.[22]. Cells were maintained at permissive temperature (33 °C) in a humidified incubator with 5% CO2 and cultured in Dulbecco's modified Eagle's medium (DMEM, Corning, 10-013), supplemented with 10% fetal bovine serum (FBS, Gemini Bio-Products, 100–106), 1% penicillin/streptomycin (Gemini Bio-Products, 400-109) and 400 µg/ml G418 (Gemini Bio-Products, 400-113). Cells were routinely tested for mycoplasma contamination using Mycoplasma PCR Detection kit (Applied Biological Materials, G238), and they

were sub-cultured at 85–90% confluency. Passage number was maintained below 14. For cell treatment experiments, the medium was removed 24 h after seeding and cells were washed once with Dulbecco's Phosphate-Buffered Saline (DPBS) solution. Serum-free medium containing either vehicle (DPBS) or 4 mM 4-deoxypyridoxine hydrochloride (DOP—Sigma Aldrich, D0501) was subsequently added for 24 h. Cells were then washed three times with ice-cold DPBS, scraped on ice and centrifuged at 500×g for 5 min at 4 °C. The pellets were flash-frozen with liquid nitrogen and stored at −80 °C until needed.

**Protein extraction and WBs**. To quantify SPL protein expression, 2 × 10⁶ striatal cells and 20 mg of frozen pulverized brain tissues (cerebral cortex) from twelve HD and eight neurologically normal controls (all male) were resuspended in 200 µl ice-cold RIPA buffer (50 mM Tris-HCl pH 7.4, 150 mM NaCl, % NP-40, 0.5 % Sodium Deoxycholate, 0.1 % SDS) supplemented with freshly made 1 mM DTT and Halt Protease and Phosphatase Inhibitor Cocktail (Thermo Scientific, 78442). Brain tissue samples were kindly provided by Prof. Richard H. Myers of Boston University. These samples were homogenized with disposable plastic pestles, while cells were vortexed for 30 s before incubation on ice for 30 min. Samples were subsequently centrifuged at 16 000×g for 10 min at 4 °C and the supernatant, containing the protein extracts, was collected. Protein concentration was measured with the Bradford Assay. WB experiments were carried out using the Odyssey infrared imaging system (Li-Cor Biosciences), as described by Ng et al.[23]. The following primary antibodies were used: anti-SGPL1 (for WB on human postmortem samples—Sigma Aldrich, SAB1408645, dilution 1:1000); anti-SGPL1 (for WB on mouse striatal cells—Abcam, ab56183, dilution 1:500); anti-Actin (Abcam, ab3280, dilution 1:10 000).

To quantify histone acetylation levels, nuclear extracts were prepared as described by Schreiber et al.[62], and 30 µg loaded on the gel for WB. The following primary antibodies were used: anti-Acetyl-Histone H3 (Lys9) (C5B11) (Cell Signaling, 9649, dilution 1:600); anti-Histone H3 (Abcam, ab10799, dilution 1:1000).

**Corticostriatal organotypic brain slice assay**. Coronal brain slices containing both cortex and striatum (250 µm thick) were prepared from Sprague-Dawley rat pups (Charles River, postnatal day 10) using a vibratome and placed into interface culture atop 0.5% agarose (JT Baker) set in Neurobasal A (Invitrogen) based culture medium[34]. All experimental procedures including the sacrificing of animals were done in accordance with NIH guidelines and under Duke IACUC approval and oversight. Brain slices were then biolistically co-transfected (Helios Gene Gun; Bio-Rad) with YFP to visualize transfected neurons together with a

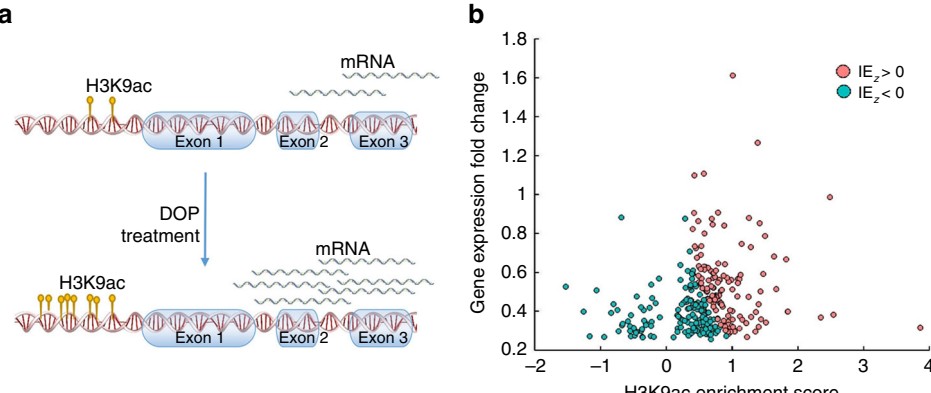

**Fig. 10** Identifying genes whose expression levels are associated with an increase in H3K9ac in the promoter. **a** Schematic representation of a gene whose expression level increases after DOP treatment, and which has an associated increase in H3K9ac level in the promoter. **b** The scatter plot shows the H3K9ac score for each gene vs. the fold change of expression levels. Each *dot* represents a gene that is upregulated after treating STHdh Q111 cells with DOP and that has at least one H3K9ac peak within 10 kb of the promoter. The *X*-axis shows the H3K9ac score for these genes. Higher scores indicate higher levels of H3K9ac after the treatment with DOP. The *Y*-axis shows the fold-change in gene expression after the DOP treatment. Genes whose expression levels are associated with an increase in H3K9ac are colored *red* (z-score of interaction effect ($IE_Z$) > 0)

mutant huntingtin expression construct for human *huntingtin* exon-1 containing a 73 CAG repeat (Httex1-Q73). DOP was added to cultures at the time of slice preparation and transfection at the indicated concentrations. After 4 days, YFP co-transfected MSNs were identified based on their location in the striatal regions of each brain slice and by their characteristic dendritic morphology, and scored as healthy if exhibiting continuous and even expression of YFP throughout the cell soma and all processes, and at least two clear primary dendrites that were at least two cell bodies long[34, 35]. MSNs exhibiting typically sized cell soma, continuous and even expression of YFP in the cell body and all processes, and at least two clear primary dendrites that were at least two cell bodies long were scored as healthy.

**Targeted lipid measurements in STHdh cell lines**. For the measurement of S1P in STHdh cell lines, metabolites were extracted from cell pellets ($3 \times 10^7$ cells) in 80% methanol containing Prostaglandin E2-d4 (PGE2-d4) as an internal standard (Cayman Chemical Co). A standard solution containing S1P (Cayman Chemical Co, 62570) was prepared in methanol and analyzed alongside the samples to match the retention time of the observed features. The LC–MS system used for the analysis comprises a Nexera X2 U-HPLC system (Shimadzu Scientific Instruments; Marlborough, MA) and a Q Exactive hybrid quadrupole orbitrap mass spectrometer (Thermo Fisher Scientific; Waltham, MA). Cell extracts and 0.4 ng of the S1P standard were injected onto a $150 \times 2$ mm ACQUITY T3 column (Waters; Milford, MA). MS analyses were carried out using electrospray ionization in the negative ion mode using full scan analysis is with an ion spray voltage of $-3.5$ kV, capillary temperature of 320 °C and probe heater temperature of 300 °C. Targeted processing and manual inspection of S1P was conducted using TraceFinder software (Thermo Fisher Scientific; Waltham, MA).

**Sphingolipid measurement from brain samples of R6/2 mice**. R6/2 transgenic mice (CAG repeat length 320–350; C57BL/6 J background, and hybrid C57BL/6×CBA (B6/CBA) background) and non-transgenic littermate controls were maintained in accordance with the IACUC policies on animal welfare at The Rockefeller University, and the ethics for the animal study was approved by The Rockefeller University Institutional Animal Care and Use Committee. Striatal tissue from the animals was rapidly dissected, placed into ice-cold lysis buffer (10 mM HEPES [pH 7.4], 150 mM KCl, 5 mM MgCl2), and disrupted by homogenization and sonication. Samples were flash frozen in liquid nitrogen and stored at −80 °C until use. For the measurement of sphingolipid species, internal standards were spiked into all samples (Avanti Polar Lipids (LM-6002); Avanti Polar Lipids, Alabaster, AL) and metabolites were then extracted in 2:1 HPLC methanol and chloroform. Samples were then placed in a bath sonicator for 30 min and incubated at 48 °C overnight. After allowing samples to reach room temperature, 1 M KOH in methanol was added to each sample and samples were sonicated for 2 h. Acetic acid was then added to all samples. For sphingoid base analysis, half of the extracted sample was centrifuged and the pellet was re-extracted in 2:1 HPLC chloroform:methanol, re-sonicated, centrifuged, and dried under nitrogen. Sphingoid bases were reconstituted in 60:40 mixture of mobile phases A and B, which consisted of 58:41:1 methanol:water:glacial acetic acid, 5 mM ammonium acetate and 99:1 methanol:glacial acetic acid, 5 mM ammonium acetate, respectively. Samples were sonicated, vortexed, and then submitted to mass spec. Samples were run on a Waters Acquity UPLC/Sciex 5500 QTrap tandem quadrapole system and assayed in positive mode according to scheduled multiple reaction monitoring for individual sphingoid bases, sphingoid-

1-phosphates. Compound separation was performed by gradient elution on a reversed phase UPLC column under gradient conditions.

**Cell viability assays**. Cell viability was assessed using a three-color fluorescence assay and high-content imaging. For DOP treatment and siRNA-mediated knockdown, 6000 and 2500 striatal cells, respectively, were seeded in sterile, black 96-well microplates. For DOP treatment, cells were incubated for 24 h in phenol red-free and serum-free DMEM, containing either vehicle (DPBS) or 4 mM DOP. For knockdown of target genes, cells were transfected for 48 h with the appropriate siRNAs. siRNA transfected cells were washed with 1× DPBS and further incubated for 24 h in serum-free DMEM. After the 24-h incubation, Calcein AM (Thermo Scientific, C3099—final concentration: 1 µg/ml), Propidium Iodide (Thermo Scientific, P3566—final concentration: 2 µg/ml) and Hoechst 333442 (Thermo Scientific, H3570—final concentration: 2 µg/ml) were added to quantify live, dead, and total cells, respectively. After 20 min incubation at 33 °C, image acquisition was carried out with a Cellomics Arrayscan Platform (Thermo Scientific). Seven fields per well were scanned at 10× magnification and quantitative analysis was performed using Cellomics proprietary algorithm for cell viability. Cell loss was calculated as the ratio of propidium iodide-positive cells to the total cell counts.

**siRNA knockdown of target genes**. 2500 STHdh Q111 cells were seeded in sterile, black 96-well microplates. After 24 h, cells were transfected with one of 50 nM Silencer Select siRNA against *Sgpl1* (Thermo Fisher, 4390771: siRNA ID # s73644), 25 nM Silencer Select siRNA against *Ahnak* (Thermo Fisher, 4390771: siRNA ID# s83166), 50 nM Silencer Select siRNA against *Tcf12* (Thermo Fisher, 4390771: siRNA ID# s74812), 50 nM Silencer Select siRNA against *Tns1* (Thermo Fisher, 4390771: siRNA ID# s75344) 50 nM Silencer Select Negative Control siRNA (Thermo Fisher, 4390843) or BLOCK-iT Alexa Fluor Red Fluorescent Control (Thermo Fisher, 465318) using the Lipofectamine RNAiMAX Transfection Reagent (Thermo Fisher, 13778030). Silencer Select Negative Control siRNA has a sequence that does not target any mouse gene transcript. After 48 h, transfection efficiency was estimated using fluorescence from cells transfected with 25 nM and 50 nM BLOCK-iT. To determine the levels of target genes after silencing, transfected cells were washed twice with DPBS, and RNA was extracted using the RNA/DNA/Protein Purification Plus Micro Kit (Norgen Biotek Corp., 51600). Reverse transcription was carried out using the Transcriptor First Strand cDNA Synthesis Kit (Roche, 04379012001) and qPCR was performed with the KAPA SYBR FAST qPCR Kit (KapaBiosystems, KK4611) and KiCqStart SYBR Green Predesigned Primers (Sigma Aldrich) for mouse *Sgpl1* (gene ID: 20397; primer pair # 1), mouse *Ahnak* (primer pair # 1), mouse *Tcf12* (gene ID: 21406; primer pair # 1), mouse *Tns1* (gene ID: 21961; primer pair # 1), and mouse *Actb* (gene ID:1146; primer pair # 1). The latter was used as internal control. Relative gene expression was calculated with the ddCt method.

**Measurement of HDAC activity**. HDAC activity was assessed using the Color de Lys HDAC Colorimetric Assay Kit (Enzo Life Sciences, BML-AK501), according to the manufacturer's specifications. Briefly, 25 µg of nuclear protein extracts from STHdh Q111 cells were incubated with 10 µM S1P (Cayman Chemical Co, 62570) in the presence of the Assay Buffer and the Color de Lys Substrate. After 15 min at 37 °C, the Color de Lys Developer was added to the samples which were further

incubated for 15 min at 37 °C. Absorbance at 405 nm was then measured using the Varioskan Flash Spectral Scanning Multimode Reader (Thermo Fisher Scientific).

To measure HDAC activity after the cell treatment with DOP, STHdh Q111 cells were incubated with either vehicle or 4 mM DOP in serum-free medium. After 24 h the cells were collected, nuclear extracts were prepared and HDAC activity was measured as described above.

**mRNA-sequencing**. RNA from striatal cell lines was extracted with the RNA/DNA/Protein Purification Kit (Norgen Biotek Corp., 23500), treated with DNase I and quantified with Nanodrop 2000 (Thermo Scientific). RNA quality and integrity were evaluated using the Fragment Analyzer (Advanced Analytical). All samples had an RNA Quality Number higher than 9.8. 1 μg of total RNA was used for library preparation. Libraries were generated using the TruSeq Stranded mRNA Library Prep Kit (Illumina, RS-122-2101), according to the manufacturer's instructions. Samples were submitted for single-end sequencing using an Illumina HiSeq 2000 platform, available at the MIT BioMicroCenter.

**mRNA-sequencing data analyses**. To analyze RNA-seq data, we first aligned raw sequencing reads to the mouse reference genome (University of California, Santa Cruz (UCSC), mm9) using TopHat 2 software[63]. We identified differentially expressed genes between DOP treated and untreated samples using Cuffdiff 2 software[64]. As a result, we identified 3097 genes with significant changes in expression levels (corrected $p < 0.01$). For each gene, we then calculated the $\log_2$ fold-change of the average for the transcript per million mapped (FPKM) reads between treated and untreated samples. Finally, we performed GO enrichment analyses on differentially expressed genes using GOrilla software[58].

**ChIP-sequencing**. Chromatin immunoprecipitation coupled with next generation sequencing (ChIP-seq) was carried out as described by Ng et al.[23]. The ChIP-grade anti-Histone H3 (acetyl K9) antibody (Abcam, ab4441) was used for immuno-precipitation. A negative control ChIP reaction was performed using normal rabbit IgG (Santa Cruz Biotechnology, sc-2027). Libraries were submitted for single-end sequencing.

**ChIP-sequencing data analyses**. Raw sequencing data were first aligned to a mouse reference genome (University of California, Santa Cruz (UCSC), mm9) using Bowtie 2 software[65]. Using GPS software[66], we then identified peaks representing regions of the genome associated with H3K9ac. We identified 30 791 peaks for STHdh samples treated with DOP and 29 496 peaks for the untreated samples. To measure the relative changes in read densities of the H3K9ac peaks, we compared detected peaks from treated and untreated conditions using MAnorm software[67]. MAnorm software normalizes peak intensities, then calculates an $M$-value for each peak that is representative of the relative peak intensity between two conditions ($q$-value $< 0.05$). To identify the nearest gene to these peaks, peak annotation was performed using ChIPseeker R package[68].

**The IE of H3K9ac on gene expression**. To detect genes whose expression levels are affected by an increase in H3K9ac, we first calculated an H3K9ac score for each gene. We identified 291 genes that are significantly upregulated following DOP treatment and with at least one annotated peak within 10 kb of their promoter. Next, we calculated an H3K9ac score for each one of these genes as a summation of $M$-values for the annotated peak located in the 10 kb region of the gene promoter.

To quantify the effect of H3K9ac on increased gene expression, we used an IE statistic. An IE measures the effect of two continuous variables on a response variable, and is defined as[69]:

$$Y = \gamma_0 + \gamma_1 X_1 + \gamma_2 X_2 + \gamma_3 X_1 X_2$$

Here, we represent the effect of H3K9ac on gene expression, $Y$, as the mediated effect of the H3K9as score ($X_1$) for each gene on the fold-change of gene expression ($X_2$) following DOP treatment. We then calculated the $z$-score of the IE (IE$_Z$), and ranked genes accordingly. High-ranked genes have expression levels that correlate to increase in H3K9ac levels. We then considered genes with IE$_Z > 0$ as primary candidates for functional analysis using GOrilla software[58].

To calculate IE$_Z$, we considered that $\gamma_0$ equals to zero since it does not affect the rank of genes based on IE$_Z$, while remaining $\gamma$ parameters are assumed equal to one ($\gamma_1 = \gamma_2 = \gamma_3 = 1$). To measure the robustness of our results based on the value of $\gamma$ parameters, we calculated IE$_Z$ of randomly selected values for $\gamma$ parameters, and identified genes for which IE$_Z > 0$. This process was repeated for 100 times. Our results demonstrate that for more than 92% of all iterations an identical set of genes was identified.

**Data Availability**. mRNA-sequencing and ChIP-sequencing data collected in this study are deposited at Gene Expression Omnibus (GEO), and are publicly available with accession number GSE98740. All relevant data are available from the authors upon request.

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

# ARTICLE

30. Menalled, L. et al. Systematic behavioral evaluation of Huntington's disease transgenic and knock-in mouse models. *Neurobiol. Dis.* **35**, 319–336 (2009).

31. Sonnino, S. & Chigorno, V. Ganglioside molecular species containing C18- and C20-sphingosine in mammalian nervous tissues and neuronal cell cultures. *Biochim. Biophys. Acta* **1469**, 63–77 (2000).

32. Zhao, L. et al. Elevation of 20-carbon long chain bases due to a mutation in serine palmitoyltransferase small subunit b results in neurodegeneration. *Proc. Natl Acad. Sci. USA* **112**, 12962–12967 (2015).

33. Lee, H. et al. 4-Deoxypyridoxine improves the viability of isolated pancreatic islets ex vivo. *Islets* **5**, 116–121 (2013).

34. Reinhart, P. H. et al. Identification of anti-inflammatory targets for Huntington's disease using a brain slice-based screening assay. *Neurobiol. Dis.* **43**, 248–256 (2011).

35. Crittenden, J. R. et al. CalDAG-GEFI down-regulation in the striatum as a neuroprotective change in Huntington's disease. *Hum. Mol. Genet.* **19**, 1756–1765 (2010).

36. Gräff, J. & Tsai, L.-H. Histone acetylation: molecular mnemonics on the chromatin. *Nat. Rev. Neurosci.* **14**, 97–111 (2013).

37. Hait, N. C. et al. Regulation of histone acetylation in the nucleus by sphingosine-1-phosphate. *Science* **325**, 1254–1257 (2009).

38. Buckley, N. J., Johnson, R., Zuccato, C., Bithell, A. & Cattaneo, E. The role of REST in transcriptional and epigenetic dysregulation in Huntington's disease. *Neurobiol. Dis.* **39**, 28–39 (2010).

39. Thomas, E. A. et al. The HDAC inhibitor 4b ameliorates the disease phenotype and transcriptional abnormalities in Huntington's disease transgenic mice. *Proc. Natl Acad. Sci. USA* **105**, 15564–15569 (2008).

40. Stack, E. C. et al. Modulation of nucleosome dynamics in Huntington's disease. *Hum. Mol. Genet.* **16**, 1164–1175 (2007).

41. Mielke, M. M. & Lyketsos, C. G. Alterations of the sphingolipid pathway in Alzheimer's disease: new biomarkers and treatment targets? *Neuromolecular Med.* **12**, 331–340 (2010).

42. Cutler, R. G., Pedersen, W. A., Camandola, S., Rothstein, J. D. & Mattson, M. P. Evidence that accumulation of ceramides and cholesterol esters mediates oxidative stress-induced death of motor neurons in amyotrophic lateral sclerosis. *Ann. Neurol.* **52**, 448–457 (2002).

43. Haughey, N. J. et al. Perturbation of sphingolipid metabolism and ceramide production in HIV-dementia. *Ann. Neurol.* **55**, 257–267 (2004).

44. Ceccom, J. et al. Reduced sphingosine kinase-1 and enhanced sphingosine 1-phosphate lyase expression demonstrate deregulated sphingosine 1-phosphate signaling in Alzheimer's disease. *Acta Neuropathol. Commun.* **2**, 12 (2014).

45. Pirhaji, L. et al. Revealing disease-associated pathways by network integration of untargeted metabolomics. *Nat. Methods* **13**, 770–776 (2016).

46. Desplats, P. A. et al. Glycolipid and ganglioside metabolism imbalances in Huntington's disease. *Neurobiol. Dis.* **27**, 265–277 (2007).

47. Di Pardo, A. et al. Ganglioside GM1 induces phosphorylation of mutant huntingtin and restores normal motor behavior in Huntington disease mice. *Proc. Natl Acad. Sci. USA.* **109**, 3528–3533 (2012).

48. Prager, B., Spampinato, S. F. & Ransohoff, R. M. Sphingosine 1-phosphate signaling at the blood-brain barrier. *Trends Mol. Med.* **21**, 354–363 (2015).

49. Pyne, S. & Pyne, N. J. Translational aspects of sphingosine 1-phosphate biology. *Trends Mol. Med.* **17**, 463–472 (2011).

50. Barrett, T. et al. NCBI GEO: archive for functional genomics data sets--update. *Nucleic Acids Res.* **41**, D991–D995 (2013).

51. Kuhn, A. et al. Mutant huntingtin's effects on striatal gene expression in mice recapitulate changes observed in human Huntington's disease brain and do not differ with mutant huntingtin length or wild-type huntingtin dosage. *Hum. Mol. Genet.* **16**, 1845–1861 (2007).

52. Hodges, A. et al. Brain gene expression correlates with changes in behavior in the R6/1 mouse model of Huntington's disease. *Genes Brain Behav.* **7**, 288–299 (2008).

53. Becanovic, K. et al. Transcriptional changes in Huntington disease identified using genome-wide expression profiling and cross-platform analysis. *Hum. Mol. Genet.* **19**, 1438–1452 (2010).

54. Irizarry, R. A. Summaries of Affymetrix GeneChip probe level data. *Nucleic Acids Res.* **31**, 15e–15e (2003).

55. Gentleman, R. C. et al. Bioconductor: open software development for computational biology and bioinformatics. *Genome Biol.* **5**, R80 (2004).

56. Bolstad, B. M., Irizarry, R. A., Astrand, M. & Speed, T. P. A comparison of normalization methods for high density oligonucleotide array data based on variance and bias. *Bioinformatics* **19**, 185–193 (2003).

57. Smyth, G. K. Linear models and empirical bayes methods for assessing differential expression in microarray experiments. *Stat. Appl. Genet. Mol. Biol.* **3**, Article3 (2004).

58. Eden, E., Navon, R., Steinfeld, I., Lipson, D. & Yakhini, Z. GOrilla: a tool for discovery and visualization of enriched GO terms in ranked gene lists. *BMC Bioinformatics* **10**, 48 (2009).

59. Ananth, C. V. & Kleinbaum, D. G. Regression models for ordinal responses: a review of methods and applications. *Int. J. Epidemiol.* **26**, 1323–1333 (1997).

60. Bender, R. & Grouven, U. Ordinal logistic regression in medical research. *J. R. Coll. Physicians Lond* **31**, 546–551 (1997).

61. Bishop, C. Pattern *Recognition and Machine Learning.* (Springer, 2006).

62. Schreiber, E., Matthias, P., Müller, M. M. & Schaffner, W. Rapid detection of octamer binding proteins with 'mini-extracts', prepared from a small number of cells. *Nucleic Acids Res.* **17**, 6419 (1989).

63. Kim, D. et al. TopHat2: accurate alignment of transcriptomes in the presence of insertions, deletions and gene fusions. *Genome Biol.* **14**, R36 (2013).

64. Trapnell, C. et al. Differential analysis of gene regulation at transcript resolution with RNA-seq. *Nat. Biotechnol.* **31**, 46–53 (2013).

65. Langmead, B. & Salzberg, S. L. Fast gapped-read alignment with Bowtie 2. *Nat. Methods* **9**, 357–359 (2012).

66. Guo, Y. et al. Discovering homotypic binding events at high spatial resolution. *Bioinformatics* **26**, 3028–3034 (2010).

67. Shao, Z., Zhang, Y., Yuan, G.-C., Orkin, S. H. & Waxman, D. J. MAnorm: a robust model for quantitative comparison of ChIP-Seq data sets. *Genome Biol.* **13**, R16 (2012).

68. Yu, G., Wang, L.-G. & He, Q.-Y. ChIPseeker: an R/Bioconductor package for ChIP peak annotation, comparison and visualization. *Bioinformatics* **31**, 2382–2383 (2015).

69. Cox, D. R. Interaction. *Int. Stat. Rev.* **52**, 1–31 (1984).

## Acknowledgements

We thank Prof. Richard H. Myers of Boston University for providing postmortem brain tissues, as well as the families of patients who donated their samples. Additionally, we thank Dr. Jeff Moore at Avanti Polar Lipids for his assistance. This work was supported by grants from National Institute of Health R01 GM089903, R01 NS089076 and R24 090963), and used computing resources funded by the National Science Foundation (DB1-0821391), and sequencing support from the National Institutes of Health (P30-ES002109) through the MIT BioMicro Center.

## Author contributions

L.P., P.M., and E.F. designed the study. L.P. and E.F. developed the computational approach, and L.P. performed the computational analyses. P.M. carried out the experimental validations. S.D. and B.T.W. performed WB experiments in the cell line model. B.T.W. assisted with the knock-down experiments. R.J.F., P.G., and M.H. performed the sphingolipid measurements in R6/2 mouse models. D.E.D. and D.C.L. performed DOP treatment experiments in the corticostriatal organotypic brain slices. J.A.-P. and C.B.C. performed targeted lipidomic experiments. L.P., P.M., and E.F. wrote the manuscript, and all the authors approved the final version.

## Additional information

**Competing interests:** The authors declare no competing financial interests.

