## [Peer Review File · Nature Communications]

Reviewers' comments:

Reviewer #1 (Remarks to the Author):

In this study, the authors use publicly accessible databases of gene expression changes from human brain autopsy material, focusing on increases or decreases in transcription that begin in grade 0 or 1 (Vonsattel system) and show monotonic increases or decreases with stage of disease. This narrows down the list of differentially transcribed genes in the human HD brain to approximately 200 that are upregulated and 600 that are downregulated, which are further ranked. From there, the authors focus on one upregulated gene -- SGPL1, which encodes sphingosine-phosphate-lyase (SPL), to study because they found a similar change in transcriptional regulation in an immortalized striatal cell line of Huntington disease - the STHdhQ111 line. The authors also note that 5 other mouse models of Huntington disease show transcriptional changes in genes regulating sphingolipid metabolism; among the 5 models at least 42% of all sphingolipid metabolism genes are dysregulated.

In the second part of the manuscript, the authors manipulate SPL activity or expression in STHdhQ111 cell lines. They show that inhibition of this enzyme or its downregulation by siRNA protects against serum starvation-induced apoptosis. They also see a correlation with upregulation of acetylation on H3K9, and about half of the genes upregulated after inhibition of SPL show a corresponding increase in H3K9 acetylation.

In all, this study demonstrates the utility of narrowing the focus on transcriptional dysregulation in Huntington disease to sets of genes that show consistent changes correlated with disease stage based on clinical data. That is a novel approach and should be of high interest in the field. However, experiments that focus on the SGPL1 gene dysregulation are less compelling, mainly because they are restricted to the STHdh immortalized striatal HD cell line. In fact, the authors do not show transcriptional dysregulation of the SGPL1 gene in HD mouse models, despite the fact that many other genes controlling sphingolipid metabolism exhibit altered transcription in those animal models.

Major points:

- 1) Experiments to alter activity or levels of SPL should be repeated in at least one HD mouse model, even if this is just in cultures of primary neurons from an appropriate HD mouse model. The STHdh cell line, while useful as a screening tool, has a distinctly different transcriptional profile from primary striatal neurons or striatal tissue. This is not surprising, given the fact that the STHdh line is immortalized and is not a mature neuronal cell type.
- 2) The impact of the work could be increased by extending studies of the role of SPL to other read-outs. Related to #1, the authors could use one of the mouse models to make primary neuronal cultures and look at effects of regulating SPL activity/levels on synaptic transmission. These experiments are important because the top gene ontology enrichment module for phenotype-associated genes in Huntington disease is synaptic transmission, as shown in Fig. 1b.
- 3) Related to # 1 and 2, the relevance of serum starvation-induced apoptosis of STHdhQ111 cells to pathogenesis of Huntington disease is not immediately clear.

Minor:

- 1) Figure 1, legend: refers to C, but there is only an A and B.
- 2) In figure 2, legend: the description of A and B are reversed. Also, in this figure and in others, there is no indication of the number of times the experiment was repeated. Please include the N for all data presented in the manuscript.

3) Page 5, top: the authors state that they use cerebral cortex tissue to assess levels of SPL because this "... allowed us to validate... model... independent of any potential variation in cell-type composition..." This statement is unlikely to be true, as pyramidal neurons in the cortex are far more sensitive to degeneration than other cell types in Huntington disease.

Reviewer #2 (Remarks to the Author):

Major comments:

Considering ordinal data in this context is innovative and important. It is true that so far this type of analysis was not applied enough but there are many examples where similar analyses were performed. These are not mentioned.

The application to a cohort of Huntington's disease studies is interesting. The computational analyses seems to be not advanced and do not go deep enough to understand the mechanisms involved.

The experimental validation of an enzyme that might be involved with the experimental validation is a nice proof of concept case study.

The consideration of labels on the samples can be done in a supervised or unsupervised manner. When done in an unsupervised way, what is the concordance? Or in other words wouldn't the samples naturally cluster by their Vonsattel grades?

Lack of background about similar computational methods with comparisons.

The ordinal regression model comes from the field of biostatistics but the problem seem to fit classical machine learning problems where non-linear methods are known to perform much better.

If you just look for genes that monotonically increase or decrease with severity, wouldn't that be the same as the linear regression model?

Other enrichment and network analyses of the up and down genes would be informative. What about mouse phenotype enrichment analysis or transcription factor enrichment analysis? Why not use more sophisticated enrichment analysis statistic that takes in account the ranks of the genes?

4,421 DEGs is too many. A more stringent cutoff should be applied.

What about the known target genes for SP1 from ChIP-seq studies? Are they differentially expressed?

It seems that the investigators already knew that this enzyme is likely involved based on their prior studies. This weakens their seemingly unbiased approach.

Increase in cell viability with the drug or with the shRNAs targeting SP1 may be general and not specific to the cell type tested.

The GO terms for the enrichment analyses are very general.

The combination of differential expression with the ChIP-seq data is powerful but the conclusions seem to fall short of a real finding.

Where the identified 146 genes fall in regards to HD, and the other gene sets previously mentioned?

"genes critical for cellular function and survival" falls short from a full explanation of what could be found there.

Minor comments:

Spell out SPL in the abstract

"Nevertheless, identification and prioritization of gene subsets that influence disease phenotypes remain challenging." Not really true, much work has been done in this area.

Since the gene name in the abstract is different from the one in the introduction it needs to be clarified that one is the gene name and the other the protein that it encodes.

"The balance between ceramide and S1P levels is a critical switch that determines the apoptotic or proliferation fate of the cells, and is necessary for maintaining appropriate levels of intermediate metabolites as well as cellular homeostasis." Reference is missing.

There is no description for (b) for Figure 1.

Overall the paper is well written and easy to follow.

Reviewer #3 (Remarks to the Author):

This study describes the development of a new approach to correlate transcriptome data with HD severity that could lead to new therapeutic approaches. However, it is not clear how their ordinal regression model can be used to integrate clinical and transcriptomic data since Figure 1a only shows a cartoon of theoretical data and Figure 1b does not show any data on gene expression and only shows HD phenotype-associated genes. It is not clear from the model how they decided to focus only on S1P lyase (Sgpl1 is only number 28 on their list of genes). There is also a lack of data showing correlations between expression of SPL or any sphingolipid metabolic enzyme or sphingolipid levels with HD disease progression. The rationale for focusing attention on SPL is weakly developed and the authors inappropriately cited a manuscript submitted to an unknown journal in Reference 27 that contains primary data providing the rationale for a major part of this study. This critical data should be included in the present manuscript.

There are a number of other important concerns outlined below.

1. Figure 1: Supplementary Table 5 shows the number of HD patient samples in four ordinal categories of disease. Gene expression data in these groups vs neurodegenerative grades should be shown instead of theoretical lines to establish the validity of the model.

Figure 2: The panel legends are switched. The antibody used for immunoblotting SPL was made against human SPL protein. There is no data in the literature or from the supplier to support its use in Panel a to detect mouse SPL. Control blots of Q111 cells after downregulation of SPL expression should be shown, molecular weight markers should be added, and the complete blots provided as supplemental data. Without verification of the specificity of the anti-SPL antibody, all of the conclusions about SPL expression in the mouse HD model cells are not valid. Panel b should show SPL expression in samples from patients within the 4 ordinal categories of HD disease described in Supplementary Table 5 to establish correlation between SPL levels and disease severity. It is not clear what samples were used for Panel b and how this data should be interpreted.

Figure 3: This cartoon is misleading since there is a lack of data correlating any changes in sphingolipids or expression of any of the sphingolipid metabolic enzymes and HD disease in this paper. Moreover, it seems inappropriate to place this "summary" figure here. It is also incorrect in places. For example, sphingolipid levels are not up or down regulated, they are increased or decreased, SphK is the correct abbreviation (genes should be SPHK1 and SPHK2), B4GALT6 encodes galactosyl transferase not glucosyl transferase, Glucosyle-Ceramide should be Glucosyl-Ceramide, missing ceramide synthase and ceramidase, important enzymes between Ceramide and Sphingosine.

Figures 4 and 5: Most of this data cannot be interpreted as showing effects of expression of mutant huntingtin since the authors did not show that the effects were different in WT Q7 cells. Why weren't effects of DOP on control cells shown? The lack of effects of FTY720 also cannot be interpreted since FTY720 inhibits sphingosine kinases and decreases S1P and it is also not known whether these cells take up FTY720 or phosphorylate it.

Figure 6: Panel a is trivial. It is not clear what Panel b shows or what the H3K9ac score is.

Figure 7: This is just a re-hash of inconclusive data presented in this paper and does not show anything but potential mechanisms. It should be deleted.

Table 1: Should also include data from Q7 cells to show whether changes in expression are dependent on expression of mutant huntingtin or not.

Supplementary data: not important for the paper.

Minor concerns:

What are DEGs? Not defined in the text.

Reviewer #4 (Remarks to the Author):

The manuscript entitled, "Identifying Novel Therapeutic Targets by Combining Transcriptional Data with Ordinal Clinical Measurements", by Pirhaji et al., presents a novel statistical model for identifying gene expression changes associated with disease severity in Huntington's disease, using previously published transcriptome-wide datasets. With the enormous volume of high-throughput datasets in the NCBI repositories, there is an essential need to mine these data in a relevant manner, which the authors have done in this manuscript. Analysis included transcriptomic data from n=38 HD caudate samples, n=32 control caudate samples, with the disease samples representing four different stages of degeneration based on Vonsattel grades. From these efforts, the authors have identified Sphingosine-1-phosphate lyase 1 (SGPL1), which encodes a key enzyme in sphingolipid metabolism, as an important gene that changes with disease severity in Huntington's disease. While the initial findings of SGPL1/SPL as an important player in disease pathology was from mining these datasets, follow up studies were carried out and validated in a striatal cell culture model of Huntington's disease.

Overall, the authors provide a novel, integrated mechanism for how abnormal regulation of this gene leads to disease pathology in Huntington's disease. This hypothesis is especially of interest, in that it incorporates dysregulated sphingolipid metabolism into an already-established disease mechanism in Huntington's disease (that is, transcriptional dysregulation and chromatin disruptions). Further, they provide new datasets, including a ChIP-seq dataset for H3K9 acetylation, and an RNA-seq dataset for 4-deoxypridoxine-treated striatal cells, to add to this

collection of integrated datasets. The manuscript is well-written and timely, and could provide a basis for using data repositories to identify important mechanisms in other diseases. Further, the statistical approaches were appropriate and robust. Aside from the comments stated below, what's missing from this paper is validation of the mechanistic link between elevated sphingosine-1-phosphate (S1P) and elevated histone acetylation. Rather, this is inferred from a past study. In that past study, which was published in Science in 2009, HeLa cells were used to show an interaction with S1P and HDACs1/2. Given that striatal cells were used in this study, the authors should show an interactions between these two important players, or show that S1P can alter HDAC activity. This would substantially strengthen the proposed mechanism of disease pathology.

Other comments:

- It is not stated where the post-mortem human brain samples were obtained from. Demographic information and disease stage should be provided for these sample. And why were only males used?

- Were any genes found that switch in the direction of regulation, that is, from up to down, or vice versa?

- In the Discussion on the bottom of pg. 7, the authors should elaborate on their results from the "typical differential analysis of gene expression data" citing references where their current findings are consistent with previous microarray findings.

- In the Discussion, it would help to summarize or outline a suggestion on how to translate this approach for other diseases. What types of scoring, or criteria etc. would be needed.

- Typo pg. 4, "regression"

Response to reviewers:

We thank the reviewers for their careful attention to our manuscript, and have provided detailed responses to their points below.

Reviewer #1 (Remarks to the Author):

In this study, the authors use publicly accessible databases of gene expression changes from human brain autopsy material, focusing on increases or decreases in transcription that begin in grade 0 or 1 (Vonsattel system) and show monotonic increases or decreases with stage of disease. This narrows down the list of differentially transcribed genes in the human HD brain to approximately 200 that are upregulated and 600 that are downregulated, which are further ranked. From there, the authors focus on one upregulated gene -- SGPL1, which encodes sphingosine-phosphate-lyase (SPL), to study because they found a similar change in transcriptional regulation in an immortalized striatal cell line of Huntington disease - the STHdhQ111 line. The authors also note that 5 other mouse models of Huntington disease show transcriptional changes in genes regulating sphingolipid metabolism; among the 5 models at least 42% of all sphingolipid metabolism genes are dysregulated.

In the second part of the manuscript, the authors manipulate SPL activity or expression in STHdhQ111 cell lines. They show that inhibition of this enzyme or its downregulation by siRNA protects against serum starvation-induced apoptosis. They also see a correlation with upregulation of acetylation on H3K9, and about half of the genes upregulated after inhibition of SPL show a corresponding increase in H3K9 acetylation.

In all, this study demonstrates the utility of narrowing the focus on transcriptional dysregulation in Huntington disease to sets of genes that show consistent changes correlated with disease stage based on clinical data. That is a novel approach and should be of high interest in the field. However, experiments that focus on the SGPL1 gene dysregulation are less compelling, mainly because they are restricted to the STHdh immortalized striatal HD cell line. In fact, the authors do not show transcriptional dysregulation of the SGPL1 gene in HD mouse models, despite the fact that many other genes controlling sphingolipid metabolism exhibit altered transcription in those animal models.

We thank the reviewer for her/his comments and interests in our methodology. We have provided additional experimental results to further show the role of SGPL1 in Huntington's disease, as described below.

Major points:

1) Experiments to alter activity or levels of SPL should be repeated in at least one HD mouse model, even if this is just in cultures of primary neurons from an appropriate HD mouse model. The STHdh cell line, while useful as a screening tool, has a distinctly different transcriptional profile from primary striatal neurons or striatal tissue. This is not surprising, given the fact that the STHdh line is immortalized and is not a mature neuronal cell type.

Yes, we thank the reviewer for his/her feedback. We have performed additional experiments in a coronal brain slice model of HD, co-transfected with a control YFP marker and a mutated exon-1 of Huntingtin gene with 73 CAG repeats (Httex1-Q73). Inhibition of SPL by 4 mM DOP shows significant neuroprotective effects. We described this finding in the revised manuscript on page 7 and in Figure 7.

2) The impact of the work could be increased by extending studies of the role of SPL to other read-outs. Related to #1, the authors could use one of the mouse models to make primary neuronal cultures and look at effects of regulating SPL activity/levels on synaptic transmission. These experiments are important because the top gene ontology enrichment module for phenotype-associated genes in Huntington disease is synaptic transmission, as shown in Fig. 1b.

In addition to cellular viability, we have measured the effect of SPL inhibition on epigenetic dysregulation, and showed a significant increase in the levels of H3K9ac. We subsequently detected genes whose upregulation is associated with increase in H3K9ac levels, and showed these genes are enriched in neuronal parts and functions (Figure 10). Experiments to analyze synaptic transmission are technically extremely challenging, and lie outside the scope of this paper.

3) Related to # 1 and 2, the relevance of serum starvation-induced apoptosis of STHdhQ111 cells to pathogenesis of Huntington disease is not immediately clear.

We thank the reviewer for this point. We provided more detailed explanation of the relevance of serum starvation-induced apoptosis on page 5.

Minor:

1) Figure 1, legend: refers to C, but there is only an A and B.

We thank the reviewer for pointing this out. We edited the figure legend accordingly.

2) In figure 2, legend: the description of A and B are reversed. Also, in this figure and in others, there is no indication of the number of times the experiment was repeated. Please include the N for all data presented in the manuscript.

We thank the reviewer for pointing this out. We edited the figure legend accordingly, and included the number of replicates in the legend.

3) Page 5, top: the authors state that they use cerebral cortex tissue to assess levels of SPL because this "... allowed us to validate... model... independent of any potential variation in cell-type composition..." This statement is unlikely to be true, as pyramidal neurons in the cortex are far more sensitive to degeneration than other cell types in Huntington disease.

We thank the reviewer for his/her comment. The reviewer is right in pointing out that pyramidal neurons are affected in HD; indeed, intracellular inclusions, transcriptional dysregulation and electrophysiological abnormalities have been described for these cells in Huntington's disease. However, while extensive degeneration of medium spiny neurons is observed in the striatum of patients, the loss of neuronal cells in cortical tissue is less severe. This was recently demonstrated by Hoss and colleagues (PLoS Genet. 2014 Feb 27;10(2):e1004188) in which cell nuclei from prefrontal cortex of 28 HD cases and 19 controls were stained with an antibody against NeuN, a neuron-specific nuclear antigen, and subsequently analyzed by flow cytometry. No differences in cell counts of the NeuN-positive fraction were observed between HD and control samples. We added this reference to the manuscript on page 5.

Reviewer #2 (Remarks to the Author):

Major comments:

- Considering ordinal data in this context is innovative and important. It is true that so far this type of analysis was not applied enough but there are many examples where similar analyses were performed. These are not mentioned.

We thank the reviewer for his/her comments about the importance of our developed methodology. We have added an extra paragraph to the introduction about existing methodologies.

- The application to a cohort of Huntington's disease studies is interesting. The computational analyses seems to be not advanced and do not go deep enough to understand the mechanisms involved.

While our methodology cannot provide detailed mechanisms, that was not our goal. Rather, we developed an easy-to-apply framework to distinguish the genes whose transcriptional regulation could have an important role in disease progression, and rank these genes. This approach provides stronger hypotheses for further experiments. Here, we show an example of this case by further validating the pathogenic effects of SPL.

- The experimental validation of an enzyme that might be involved with the experimental validation is a nice proof of concept case study.

We thank the reviewer for his/her comment.

-The consideration of labels on the samples can be done in a supervised or unsupervised manner. When done in an unsupervised way, what is the concordance? Or in other words wouldn't the samples naturally cluster by their Vonsattel grades?

Yes, both supervised and unsupervised approaches can be considered for clustering the samples. Hierarchical clustering yields coarse clusters containing some of the samples with the same grade as shown in the following figure. However, the goal of our approach is not to identify the clusters of samples, but rather to discover genes whose expression is significantly associated with disease progression.

- OC-1: Ordinal Category 1
- OC-2: Ordinal Category 2
- OC-3: Ordinal Category 3
- OC-4: Ordinal Category 4

- Lack of background about similar computational methods with comparisons.

We thank the reviewer for pointing this out. As mentioned above, we have added an extra paragraph to the methods regarding the existing computational methods.

-The ordinal regression model comes from the field of biostatistics but the problem seem to fit classical machine learning problems where non-linear methods are known to perform much better.

If you just look for genes that monotonically increase or decrease with severity, wouldn't that be the same as the linear regression model?

Although a linear regression model can be considered for detecting monotonic increases or decreases in the severity of the disease with quantitative values, it cannot consider the qualitative ordering of disease severity. In contrast, as described in the manuscript, the ordinal regression model can be applied in cases where disease progression is described qualitatively such as many clinical scales. For examples, a Vonsattel score of four does not represent twice the severity of a score of two. Therefore, an ordinal regression model can deal with these type of data, since it makes no assumptions about the relative quantitative value of the scale. We have explained this concept more clearly in the revised manuscript by modifying Figure 1 and adding additional explanation.

Furthermore, the DAGs identified by our ordinal regression model differ from the monotonically changing genes. We identified 1,688 genes whose expression levels monotonically increase or decrease with the progression of neurodegeneration. Only 267 (<16%) of these genes overlap with the phenotype-associated (PAG) genes identified by the ordinal regression model. Notably, the top ranked PAGs, *BCL2L11* and *SPI*, as well as *SGPL1* are not in the list of monotonically changing genes.

- Other enrichment and network analyses of the up and down genes would be informative. What about mouse phenotype enrichment analysis or transcription factor enrichment analysis? Why not use more sophisticated enrichment analysis statistic that takes in account the ranks of the genes?

The reviewer is correct that there are many ways to mine an expression dataset. However, the scope of our manuscript is to identify a subset of the differentially expressed genes that may have a role in disease progression that can be considered for therapy. Therefore, we followed up with the top ranked gene for further experimental validations.

- 4,421 DEGs is too many. A more stringent cutoff should be applied.

Yes, we agree with the reviewer that 4,421 genes are too many; however, they are identified by a stringent cutoff of corrected p-value < 0.001. Addressing this problem is one of the motivations behind developing our methodology. As shown in Fig. 2a, using an ordinal regression model we can detect a small subset of these genes (<20%), which represent the majority of biological processes inferred from all the DEGs.

- What about the known target genes for SP1 from ChIP-seq studies? Are they differentially expressed?

That is a very interesting question. However, because of the scope of our work, we did not focus on more detailed analysis of SP1 targets.

- It seems that the investigators already knew that this enzyme is likely involved based on their prior studies. This weakens their seemingly unbiased approach.

We thank the reviewer for pointing this out. To further validate our unbiased approach, we have now assessed cellular viability after knockdown of four of the top-ranked genes. Our results showed 75% of these genes are modifiers of cellular viability. These new results and data are shown as Figure 3 and described on page 5.

- Increase in cell viability with the drug or with the shRNAs targeting SP1 may be general and not specific to the cell type tested.

We thank the reviewer for pointing this out. To address this concern, we measure the effect of SPL inhibition by DOP on the normal control cells. The results are shown in Figure 6b and 6c, and described on page 6.

- The GO terms for the enrichment analyses are very general.

Yes, although some of the GO terms refer to more general biological processes, we have provided many specific GO terms such as Ras guanyl-nucleotide exchange factor activity, response to interferon-beta, regulation of axonogenesis, and synaptic transmission.

- The combination of differential expression with the ChIP-seq data is powerful but the conclusions seem to fall short of a real finding.

Clearly, the standard for a “real finding” is a subjective one. As reviewer #4 noted, we have linked “dysregulated sphingolipid metabolism into an already-established disease mechanism in Huntington's disease (that is, transcriptional dysregulation and chromatin disruptions)”. In the revised manuscript we provide further evidence of this link by showing that treatment of the HD striatal cells with an inhibitor of the enzyme reduces HDAC activity.

Where the identified 146 genes fall in regards to HD, and the other gene sets previously mentioned?

"genes critical for cellular function and survival" falls short from a full explanation of what could be found there.

We thank the reviewer for raising this concern. Yes, we agree with the reviewer about this point and removed this claim from our manuscript.

Minor comments:

- Spell out SPL in the abstract

Yes, we edited the manuscript accordingly.

- "Nevertheless, identification and prioritization of gene subsets that influence disease phenotypes remain challenging." Not really true, much work has been done in this area.

Yes, we believe there is much work done in this field, we have described the problems with current approaches in an additional paragraph in the introduction.

-Since the gene name in the abstract is different from the one in the introduction it needs to be clarified that one is the gene name and the other the protein that it encodes.

Yes, we edited the manuscript accordingly.

- "The balance between ceramide and SIP levels is a critical switch that determines the apoptotic or proliferation fate of the cells, and is necessary for maintaining appropriate levels of intermediate metabolites as well as cellular homeostasis." Reference is missing.

We thank the reviewer for pointing this out, we edited the manuscript accordingly.

- There is no description for (b) for Figure 1.

We thank the reviewer for pointing this out. We have edited the figure legend panel names.

- Overall the paper is well written and easy to follow.

We thank the reviewer for his/her comments about our manuscript.

Reviewer #3 (Remarks to the Author):

This study describes the development of a new approach to correlate transcriptome data with HD severity that could lead to new therapeutic approaches. However, it is not clear how their ordinal regression model can be used to integrate clinical and transcriptomic data since Figure 1a only shows a cartoon of theoretical data and Figure 1b does not show any data on gene expression and only shows HD phenotype-associated genes. It is not clear from the model how they decided to focus only on S1P lyase (Sgpl1 is only number 28 on their list of genes). There is also a lack of data showing correlations between expression of SPL or any sphingolipid metabolic enzyme or sphingolipid levels with HD disease progression. The rationale for focusing attention on SPL is weakly developed and the authors inappropriately cited a manuscript submitted to an unknown journal in Reference 27 that contains primary data providing the rationale for a major part of this study. This critical data should be included in the present manuscript.

We thank the reviewer for his/her comments, and apologize for the lack of clarity in justification of our methodology. To address these concerns, we have edited the manuscript and provided additional data:

- We have modified Figure 1, added additional supplementary Figure 2 and 3 to show the actual data points, and explained the methods more clearly.
- We have performed additional experiments to validate the top ranked genes identified by our method.
- We showed that the levels of S1P, which is the substrate of the SPL enzyme, are significantly decreased with the progression of HD in R6/2 mouse model of HD.
- The rationale behind selection of SPL is now more clearly explained. It was the highest ranked gene from the human data that was also differentially expressed the murine HD samples that we used for our experimental tests.
- We apologize for ambiguity on the reference 27. That article is published and we have now cited it appropriately.

There are a number of other important concerns outlined below.

- Figure 1: Supplementary Table 5 shows the number of HD patient samples in four ordinal categories of disease. Gene expression data in these groups vs neurodegenerative grades should be shown instead of theoretical lines to establish the validity of the model.

Yes, we agree with the reviewer in the value of showing the real data, and we have now provided the gene expression data plots as supplementary figure 2 and 3. We continue to use theoretical lines in Figure 1 to make the explanation of our approach more clear.

- Figure 2: The panel legends are switched. The antibody used for immunoblotting SPL was made against human SPL protein. There is no data in the literature or from the supplier to support its use in Panel a to detect mouse SPL. Control blots of Q111 cells after downregulation of SPL expression should be shown, molecular weight markers should be added, and the complete blots provided as supplemental data. Without verification of the specificity of the anti-SPL antibody, all of the conclusions about SPL expression in the mouse HD model cells are not valid. Panel b should show SPL expression in samples from patients within the 4 ordinal categories of HD disease described in Supplementary Table 5 to establish correlation between SPL levels and disease severity. It is not clear what samples were used for Panel b and how this data should be interpreted.

We thank the reviewer for pointing this out. We have edited the figure legend accordingly. With regards to SPL Western Blotting experiments, we used two different antibodies for mouse and human extracts, as we described in the “Materials and Methods” section. Specifically, for mouse extracts we used this antibody: Abcam, ab56183. According to the manufacturer’s datasheet, this antibody reacts with SPL protein from mouse and rat. We show the complete WB below. Furthermore, we have provided SPL WB results after *Sgpl1* knock-down in STHdh Q111 cells in Supplementary Fig 5.

We thank the reviewer for his/her comment about the human samples. As explained in the manuscript, we chose cortical tissue because we wanted to validate the ability of the proportional odds model in detecting HD-related genes independent of significant variations in cell-type composition. In fact, our statistical approach requires that the genes be altered even in low-severity samples, without extensive neuronal loss in the striatum, and it requires that the expression of the gene must consistently increase or decrease with HD severity.

Cortical involvement in HD is well known: pyramidal cells in the cortex presents intracellular aggregation and show widespread changes in gene expression. Nevertheless, this tissue does not experience dramatic neuronal loss and related alteration in cell-type composition. Therefore, we selected this tissue for our validation.

- Figure 3: This cartoon is misleading since there is a lack of data correlating any changes in sphingolipids or expression of any of the sphingolipid metabolic enzymes and HD disease in this paper. Moreover, it seems inappropriate to place this "summary" figure here. It is also incorrect in places. For example, sphingolipid levels are not up or down regulated, they are increased or decreased, SphK is the correct abbreviation (genes should be SPHK1 and SPHK2), B4GALT6 encodes galactosyl transferase not glucosyl transferase, Glucosyle-Ceramide should be Glucosyl-Ceramide, missing ceramide synthase and ceramidase, important enzymes between Ceramide and Sphingosine.

We thank the reviewer for his/her concerns about this figure. We agree with the reviewer on this point and removed this figure in the revised manuscript.

- Figures 4 and 5: Most of this data cannot be interpreted as showing effects of expression of mutant huntingtin since the authors did not show that the effects were different in WT Q7 cells. Why weren't effects of DOP on control cells shown? The lack of effects of FTY720 also cannot be interpreted since FTY720 inhibits sphingosine kinases and decreases S1P and it is also not known whether these cells take up FTY720 or phosphorylate it.

-Yes, we agree with the reviewer's concerns. To address them, we have provided additional experimental data about the effect of DOP on control cell (Figure 8).

Additionally, we agree with the concerns about FTY720 results and have therefore removed these data from the manuscript.

- Figure 6: Panel a is trivial. It is not clear what Panel b shows or what the H3K9ac score is. We thank the reviewer for his/her feedback. We believe panel a provides a simplified explanation of the selection of those genes whose increase in expression is significantly associated with increase in H3K9ac levels. We have provided additional explanation about H3K9ac score.

- Figure 7: This is just a re-hash of inconclusive data presented in this paper and does not show anything but potential mechanisms. It should be deleted.

We agree with the reviewers, and removed Figure 7 in the revised manuscript.

- Table 1: Should also include data from Q7 cells to show whether changes in expression are dependent on expression of mutant huntingtin or not.

Yes, we agree with the reviewer and provided a hierarchical clustering in Figure 9 that shows gene expression levels of STHdh Q7 cells, STHdh Q111 cells, and STHdh Q111 cells after treatment with DOP. We further identified the cluster of genes for which DOP treatment caused their expression in STHdh Q111 to be corrected in the direction of the expression level in STHdh Q7 cells.

- Supplementary data: not important for the paper.

We have modified the supplementary materials to show the new data and results.

Minor concerns:

What are DEGs? Not defined in the text.

We thank the reviewer for pointing this out. DEGs refers to differentially expressed genes, we wrote out the phrase before using the acronym in the revised manuscript.

Reviewer #4 (Remarks to the Author):

The manuscript entitles, "Identifying Novel Therapeutic Targets by Combining Transcriptional Data with Ordinal Clinical Measurements", by Pirhaji et al., presents a novel statistical model for identifying gene expression changes associated with disease severity in Huntington's disease, using previously published transcriptome-wide datasets. With the enormous volume of high-throughput datasets in the NCBI repositories, there is an essential need to mine these data in a relevant manner, which the authors have done in this manuscript. Analysis included transcriptomic data from n=38 HD caudate samples, n=32 control caudate samples, with the disease samples representing four different stages of degeneration based on Vonsattel grades. From these efforts, the authors have identified Sphingosine-1-phosphate lyase 1 (SGPL1), which encodes a key enzyme in sphingolipid metabolism, as an important gene that changes with disease severity in Huntington's disease. While the initial findings of SGPL1/SPL as an important player in disease pathology was from mining these datasets, follow up studies were carried out and validated in a striatal cell culture model of Huntington's disease.

We thank the reviewer for his/her comments on our manuscript.

Overall, the authors provide a novel, integrated mechanism for how abnormal regulation of this gene leads to disease pathology in Huntington's disease. This hypothesis is especially of interest, in that it incorporates dysregulated sphingolipid metabolism into an already-established disease mechanism in Huntington's disease (that is, transcriptional dysregulation and chromatin disruptions). Further, they provide new datasets, including a ChIP-seq dataset for H3K9 acetylation, and an RNA-seq dataset for 4-deoxy pyridoxine-treated striatal cells, to add to this collection of integrated datasets. The manuscript is well-written and timely, and could provide a basis for using data repositories to identify important mechanisms in other diseases. Further, the statistical approaches were appropriate and robust. Aside from the comments stated below, what's missing from this paper is validation of the mechanistic link between elevated sphingosine-1-phosphate (S1P) and elevated histone

acetylation. Rather, this is inferred from a past study. In that past study, which was published in Science in 2009, HeLa cells were used to show an interaction with S1P and HDACs1/2. Given that striatal cells were used in this study, the authors should show an interactions between these two important players, or show that S1P can alter HDAC activity. This would substantially strengthen the proposed mechanism of disease pathology.

We appreciate reviews comments about the novelty of our approach and our findings.

Yes, we agree with the reviewer about providing validation of the link between S1P levels and changes in histone acetylation. To address this concern we performed additional experiments; the results validate that S1P is a modulator of HDAC activity in our cell line model and show that the treatment of the mutated striatal cells with DOP diminishes HDAC activity. These new results are described on page 7 and shown in Figures 8b and 8c.

Other comments:

- It is not stated where the post-mortem human brain samples were obtained from. Demographic information and disease stage should be provided for these sample. And why were only males used?

We thank the review for pointing this out. The samples were provided by Prof. Richard H. Myers of Boston University; we included this information in the “Materials and Methods” section. Furthermore, we provided additional information about the samples in the Supplementary Table 3. We used only samples from male HD and control subjects to avoid any potential bias associated with sex-related factors.

- Were any genes found that switch in the direction of regulation, that is, from up to down, or vice versa?

This is a very interesting question. To answer this question we added an additional Figure (Figure 9), which shows the expression levels of differentially expressed genes in STHdh Q7, STHdh Q111 and STHdh Q111 treated with DOP. We found three clusters of genes in STHdh Q111 whose expression was corrected toward wild-type expression when treated with DOP.

- In the Discussion on the bottom of pg. 7, the authors should elaborate on their results from the "typical differential analysis of gene expression data" citing references where their current findings are consistent with previous microarray findings.

We appreciate that the reviewer raised this point, we edited the revised manuscript to make this point more clear. Additionally, we added the supplementary Figure 1 that compares our findings with previous findings. In this figure, we show not only that our approach can discover the

previously reported findings, it also infers higher enrichment scores for the previously reported dysregulated processes in HD.

- In the Discussion, it would help to summarize or outline a suggestion on how to translate this approach for other diseases. What types of scoring, or criteria etc. would be needed.

Yes, we have added additional explanation

- Typo pg. 4, "regression"

We thank the reviewer for pointing this out. We edited accordingly in the revised manuscript.

Reviewers' comments:

Reviewer #3 (Remarks to the Author):

This is a greatly improved version of this manuscript and the authors adequately addressed most of my concerns. However, there are still two issues that need to be corrected.

1. The authors have mistakes in all of the sphingolipid nomenclature shown in the mass spectrometry data. However, their results could be accurate showing decreases in all phosphorylated sphingoid bases as well as the sphingoid bases. Naturally occurring S1P is d18:1 sphingosine-1-phosphate. The authors should check the Avanti catalog for correct nomenclature of phosphorylated sphingoid bases as well as the sphingoid bases. Supplementary Figure 7 – names of all sphingoid bases should be corrected and all data on phosphorylated sphingoid bases should be shown in the main figures. They should also carefully check their mass spectrometry data to make sure that they have identified the correct molecular ions. This is an important issue that needs to be corrected.

2. Supplementary Figure 5A is important to show that the antibody for SPL is indeed specific and should be included with Figure 4.

Reviewer #4 (Remarks to the Author):

The authors present an extensively revised manuscript that is much improved over the initial paper. The resulting manuscript is comprehensive, well-written, interesting and timely. This reviewer has no additional comments, except for the minor changes below.

- Spell out gene names the first time they appear. For example, SGPL1, BCL2L11, Ahnak, Tcf12 and Tns1.

- Supplementary Table 4 should have gene names spelled out.

Response to reviewers:

We thank the reviewers for their careful attention to our manuscript, and pointing out the improvements in the revised manuscript. We have provided detailed responses to their points below.

Reviewer #3 (Remarks to the Author):

This is a greatly improved version of this manuscript and the authors adequately addressed most of my concerns. However, there are still two issues that need to be corrected.

We thank the reviewer for comments on our manuscript and its improvements.

1. However, their results could be accurate showing decreases in all phosphorylated sphingoid bases as well as the sphingoid bases.

Naturally occurring S1P is d18:1 sphingosine-1-phosphate.

The authors should check the Avanti catalog for correct nomenclature of phosphorylated sphingoid bases as well as the sphingoid bases.

Supplementary Figure 7 – names of all sphingoid bases should be corrected and all data on phosphorylated sphingoid bases should be shown in the main figures. They should also carefully check their mass spectrometry data to make sure that they have identified the correct molecular ions. This is an important issue that needs to be corrected.

We thank the reviewer for raising this important point regarding to the nomenclature of the measured metabolites. After careful investigation of our original mass spectrometry data, we found several typographical errors in the name of sphingoid bases. We have corrected them accordingly in the revised manuscript and the corresponding supplementary figure.

We apologize for the lack of clarity in the identified phosphorylated sphingoid bases. We found that d18:1 S1P is significantly decreased in 6-week-old R6/2 mice compared to the corresponding controls (p-value=0.03, Fig. 5a). In addition to sphingolipids with a carbon length of 18, which include the main components of cellular sphingolipids (Sonnino & Chigorno, 2000), we detected changes in sphingolipids with a carbon length of 20. Interestingly, the detection of sphingolipids with a carbon length of 20 has been reported in brain and central nervous systems (Sonnino & Chigorno, 2000, Pruett et al., 2008, Maceyka, Milstien, & Spiegel, 2005), and their dysregulation has been associated with neurodegeneration (Zhao et al., 2015). We determined that the levels of d20:1 S1P and total phosphorylated sphingoid bases are significantly decreased in both 6-week (Fig. 5b and 5c respectively) and 22-week (Fig. 5d and 5e respectively) R6/2 mice compared to their corresponding controls.

Following the reviewer's suggestions, we added these findings about phosphorylated sphingoid bases to the main Figure 5.

2. Supplementary Figure 5A is important to show that the antibody for SPL is indeed specific and should be included with Figure 4.

Yes, we thank the reviewer for pointing this out. As suggested, we included the result of the siRNA-mediated knock-down experiment proving the specificity of the SPL antibody (Abcam, ab56183) in Figure 4.

Reviewer #4 (Remarks to the Author):

The authors present an extensively revised manuscript that is much improved over the initial paper. The resulting manuscript is comprehensive, well-written, interesting and timely. This reviewer has no additional comments, except for the minor changes below.

We thank the reviewer for his/her positive comments on our revised manuscript.

- Spell out gene names the first time they appear. For example, SGPL1, BCL2L11, Ahnak, Tcf12 and Tns1.

- Supplementary Table 4 should have gene names spelled out.

We thank the reviewer for pointing this out. We now spell out the genes when then first appear in the manuscript, and Supplementary Table 4 is edited accordingly.

References:

- Maceyka, M., Milstien, S., & Spiegel, S. (2005). Sphingosine kinases, sphingosine-1-phosphate and sphingolipidomics. *Prostaglandins & Other Lipid Mediators*, 77(1–4), 15–22.
<http://doi.org/10.1016/j.prostaglandins.2004.09.010>
- Pruett, S. T., Bushnev, A., Hagedorn, K., Adiga, M., Haynes, C. A., Sullards, M. C., ... Merrill, A. H. (2008). Thematic Review Series: Sphingolipids. Biodiversity of sphingoid bases ("sphingosines") and related amino alcohols. *The Journal of Lipid Research*, 49(8), 1621–1639. <http://doi.org/10.1194/jlr.R800012-JLR200>
- Sonnino, S., & Chigorno, V. (2000). Ganglioside molecular species containing C18- and C20-sphingosine in mammalian nervous tissues and neuronal cell cultures. *Biochimica et Biophysica Acta (BBA) - Reviews on Biomembranes*, 1469(2), 63–77.
[http://doi.org/10.1016/S0005-2736\(00\)00210-8](http://doi.org/10.1016/S0005-2736(00)00210-8)
- Zhao, L., Spassieva, S., Gable, K., Gupta, S. D., Shi, L.-Y., Wang, J., ... Nishina, P. M. (2015). Elevation of 20-carbon long chain bases due to a mutation in serine palmitoyltransferase small subunit b results in neurodegeneration. *Proceedings of the National Academy of Sciences of the United States of America*, 112(42), 12962–7.
<http://doi.org/10.1073/pnas.1516733112>

REVIEWERS' COMMENTS:

Reviewer #3 (Remarks to the Author):

The authors adequately addressed all previous concerns